# Crystallographic and spectroscopic assignment of the proton transfer pathway in [FeFe]-hydrogenases

Jifu Duan [1], Moritz Senger[2], Julian Esselborn[1], Vera Engelbrecht[1], Florian Wittkamp[3], Ulf-Peter Apfel[3,4], Eckhard Hofmann [5], Sven T. Stripp [2], Thomas Happe[1] & Martin Winkler[1]

The unmatched catalytic turnover rates of [FeFe]-hydrogenases require an exceptionally efficient proton-transfer (PT) pathway to shuttle protons as substrates or products between bulk water and catalytic center. For clostridial [FeFe]-hydrogenase CpI such a pathway has been proposed and analyzed, but mainly on a theoretical basis. Here, eleven enzyme variants of two different [FeFe]-hydrogenases (CpI and HydA1) with substitutions in the presumptive PT-pathway are examined kinetically, spectroscopically, and crystallographically to provide solid experimental proof for its role in hydrogen-turnover. Targeting key residues of the PT-pathway by site directed mutagenesis significantly alters the pH-activity profile of these variants and in presence of $H_2$ their cofactor is trapped in an intermediate state indicative of precluded proton-transfer. Furthermore, crystal structures coherently explain the individual levels of residual activity, demonstrating e.g. how trapped $H_2O$ molecules rescue the interrupted PT-pathway. These features provide conclusive evidence that the targeted positions are indeed vital for catalytic proton-transfer.

[1] Department of Plant Biochemistry, Photobiotechnology, Ruhr-Universität Bochum, 44801 Bochum, Germany. [2] Department of Physics, Experimental Molecular Biophysics, Freie Universität Berlin, 14195 Berlin, Germany. [3] Department of Chemistry and Biochemistry, Inorganic Chemistry I, Ruhr-Universität Bochum, 44801 Bochum, Germany. [4] Fraunhofer UMSICHT, Osterfelder Straße, 346047 Oberhausen, Germany. [5] Department of Biophysics, Protein Crystallography, Ruhr-Universität Bochum, 44801 Bochum, Germany. Correspondence and requests for materials should be addressed to T.H. (email: thomas.happe@rub.de) or to M.W. (email: martin.winkler-2@rub.de)

[F]eFe]-hydrogenases represent one of natures' most effective classes of redox enzymes catalyzing the reversible reduction of protons to dihydrogen ($H_2$) at turnover frequencies of up to 9000 s$^{-1}$[1–3]. Most [FeFe]-hydrogenases favor proton reduction while [NiFe]-hydrogenases are usually more biased toward $H_2$ oxidation[4]. With their low catalytic overpotential[5], [FeFe]-hydrogenases represent excellent models for a regenerative and likewise economically feasible $H_2$ production. Their active center ("H-cluster") can be structured into a standard [4Fe-4S]-cluster ([4Fe]$_H$) and a diiron site ([2Fe]$_H$). The latter is uniquely coordinated by three carbonmonoxide (CO) and two cyanide (CN$^-$) ligands. They stabilize the cofactor in its protein environment[6] and fine-tune its redox features[7]. An azadithiolate ligand (adt) further bridges the proximal (Fe$_p$) and the distal (Fe$_d$) iron center, which are differentiated according to their location relative to the [4Fe]$_H$-cluster.

To achieve the extraordinary high turnover frequencies of [FeFe]-hydrogenases[8], it can be implied that proton transfer (PT) is facilitated by distinct and optimized pathways. PT pathways span large distances through protein scaffolds, e.g. to enable proton-coupled electron transfer or proton translocation[9–12]. They usually comprise of a succession of protonatable or polar residues and protein-bound water molecules aligned at hydrogen-bonding distance[13,14].

Based on the crystal structures of [FeFe]-hydrogenases CpI from *Clostridium pasteurianum* and DdH from *Desulfovibrio desulfuricans*, several putative PT pathways have been discussed[15–18]. Theoretical studies suggest that the most probable PT pathway comprises of strictly conserved residues E282, S319, E279, and C299 (from surface to H-cluster, numbering corresponds to CpI), including two protein-bound water molecules (Wat826, Wat1120 4XDC[19], chain B) located between E279 and C299[15,16,20]. This pathway ends with C299 located in hydrogen-bonding distance to the amine head-group of the adt-ligand. Its identity and importance as a proton relay was verified in comparative studies on cofactor variants of HydA1 from *Chlamydomonas reinhardtii*[21]. Cornish and co-workers could show that amino-acid substitutions along the putative PT pathway dramatically decreased the catalytic activities. In particular, their study indicated a participation of surface-exposed residue E282 in catalytic PT[15]. In a second study they suggested a regulative function for positions R286 and S320 in the PT of CpI[22]. Furthermore, Morra and co-workers described that the pH optimum of variant C298D of [FeFe]-hydrogenase CaI from *Clostridium acetobutylicum* (corresponding to C299D in CpI) is shifted from pH 8 to pH 7, indicative for the involvement of C298 in PT[23,24].

Although several studies were conducted, immediate experimental evidence for the relevance of residues in the putative PT pathway is missing, leaving an essential aspect of enzymatic performance opaque. In this study, site-directed mutagenesis (SDM) is used to investigate the PT pathway of two [FeFe]-hydrogenases, namely CpI and HydA1, which represent the largest (M3) and smallest (M1) type of monomeric [FeFe]-hydrogenases, respectively[25]. Most of the 22 SDM variants show strongly affected $H_2$ release activities and pH optima. For 11 of these variants the crystal structure is solved, which facilitates a correlation of individual structural features (i.e. hydrogen-bonding distances) and catalytic performance. This provides insight into the minutiae of PT on the molecular level. Catalytically hampered SDM variants are analyzed by in situ attenuated total reflection Fourier transform infrared (ATR-FTIR) spectroscopy. When flushed with $H_2$, these enzymes are found to adopt a key intermediate of hydrogen turnover, the recently described H$_{hyd}$ state[26–30]. H$_{hyd}$ accumulation under $H_2$ clearly correlates with the diminished PT efficiency of the enzyme[29]. Herein we provide complementary kinetic, structural, and spectroscopic data, which allow to verify the PT pathway discussed above as the key route of catalytic PT in [FeFe]-hydrogenases.

## Results

**$H_2$ release assays and pH-dependent enzyme activities.** For both CpI and HydA1, 11 SDM variants were generated to target residues along the putative PT pathway applying conservative and non-conservative exchanges (Fig. 1). Conservative exchanges (e.g. E → D) maintain the functional group of the targeted position, but due to other structural differences in the substitute residue, will affect the precise spatial placement and configuration of the functional group. In a highly ordered system such as the well-distanced H-bond chain of an evolutionarily optimized PT pathway, this should at least affect the efficiency of the functional aspect. Non-conservative exchanges (e.g. E → Q/A) delete the functional group entirely and therefore prohibit these substitute residues from rescuing the targeted function. For wild-type (wt) enzyme, $H_2$ release activities of about 860 (HydA1) and 2600 (CpI) μmol $H_2$ per mg per min were measured and defined as 100% activity[29,31–33]. Corresponding substitutions in CpI and HydA1 had similar effects on $H_2$-release activity, except for variants E282Q$_{CpI}$ (8.2%) and E144Q$_{HydA1}$ (0.4%). The strongest impacts were achieved when replacing E279$_{CpI}$ for A or Q, or C299$_{CpI}$ for A or S, resulting in activities <1%. This is well in line with the overall trend of an increasing impact of substitutions along the PT pathway in the direction from surface to H-cluster with C299D being the only exception (Fig. 1b). In general, conservative amino-acid exchanges had less dramatic effects than non-conservative substitutions. However, in some cases a non-conservative exchange to alanine retained a surprisingly large fraction of activity. Variants E282D$_{CpI}$, C299D$_{CpI}$, R286A$_{CpI}$, and E282A$_{CpI}$ were only mildly affected and showed residual activities between 30 and 90%. For the same PT pathway position, different substitutions can have a dramatically different impact as exemplified by position E144$_{HydA1}$/E282$_{CpI}$. Here the exchange to glutamine diminished enzyme activity to only 0.4–8%, while the non-conservative exchange to alanine and the conservative exchange to aspartic acid retained between 46 and 81% activity. Further, there is a general trend for substitutions of the more surface-exposed PTP positions (R286, E282, and even S319) in CpI to have slightly lower impact on enzymatic activity as compared to the corresponding HydA1 variants. It might suggest for CpI a more open or flexible access to the PT pathway, which can tolerate variations slightly better than HydA1. This is especially obvious in the light of the 20-fold difference in the relative activity of E282Q$_{CpI}$, as compared to its HydA1 counterpart E144Q.

To link the role of individual residues in the putative PT pathway to substrate/product (H$^+$) transfer, we probed the pH-activity profiles of variants with a high enough level of residual activity in terms of $H_2$-release and $H_2$-oxidation activity. As shown in Fig. 2a, native HydA1 is most active from pH 7 to pH 8 while wild-type CpI (Fig. 2b) clearly reaches its highest activity at about pH 8. Variant CpI-Y572A served as a negative control, with an amino-acid substitution outside of the putative PT pathway and features the pH-dependency profile of wild-type enzyme (see Supplementary Fig. 2a). In contrast to this behavior, the pH optima of all variants targeting the putative PT pathway were shifted to lower pH values, indicating that the limited PT efficiency can at least be partially rescued by an increased proton concentration.

Just as observed for catalytic activity, different variants of the same position in the putative PT pathway can cause significantly deviating shifts in pH optimum. In case of E282$_{CpI}$, a substitution for aspartic acid shifted the optimum only slightly to pH 7.5 while

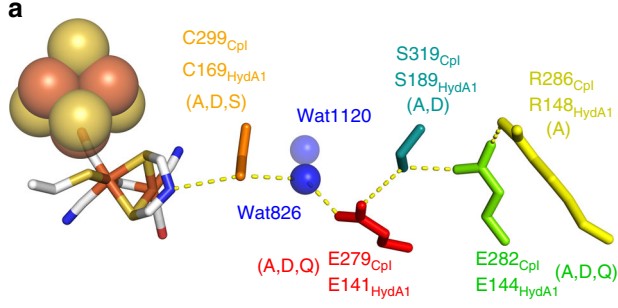

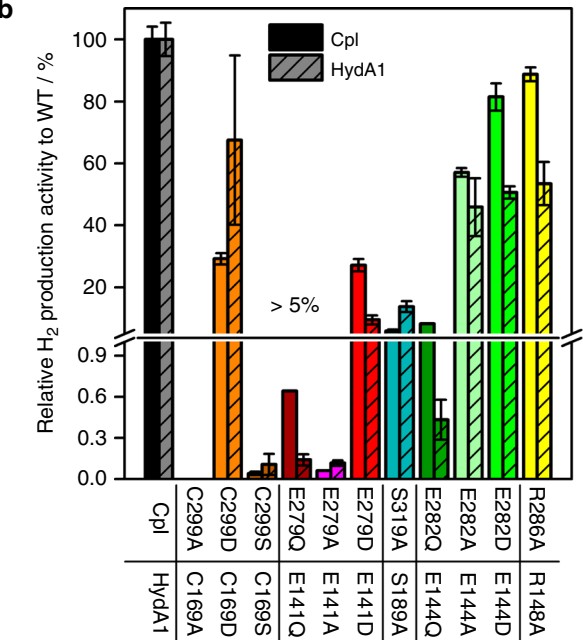

**Fig. 1** Putative PT pathway in CpI and $H_2$ evolution activities of SDM variants. **a** The PT pathway of CpI (PDB: 4XDC[19]) is presented as stick structure with individually colored residues while the [4Fe-4S]-cluster and water molecules are shown as spheres. The substitutions applied in this study for individual positions are shown in parentheses below the respective position labels. **b** $H_2$-production activity of SDM variants targeting the putative PT pathway in CpI and HydA1. $H_2$ production activities of PT pathway variants determined at pH 6.8 are presented in % relative to the respective wild-type activity. Activity bars for different variants of the same position exhibit corresponding basic colors but different shades. Relative activities higher than 5% are shown in the upper part of the discontinuous scale. Wild-type CpI and HydA1 exhibit activities of 2576 ± 107 and 862 ± 46.5 μmol $H_2$ per mg per min, respectively. The bars represent mean values from at least three independent measurements, including standard deviations. Details on the in vitro assay are presented in Supplementary Fig. 1

an exchange to alanine or glutamine rendered the enzyme most active at pH 7 and 6, respectively (Fig. 2b). In most cases, the extent of the shift correlates to an overall loss in $H_2$ release activity, however this has not been observed in all cases. The conservative variant E279D retains 10–30% of $H_2$ release activity but exhibits a significantly stronger down-shift in pH optimum than the largely inactive variant E279Q (Fig. 2c). As shown in Fig. 2 and Supplementary Fig. 2, no significant differences between CpI and HydA1 and their respective SDM variants were observed, underlining the universal impact of the putative PT pathway. To probe the relevance of the putative PT pathway for catalytic proton release, we also investigated the pH-dependency of $H_2$ oxidation. It is known from electrochemistry experiments

with different [FeFe]-hydrogenases that the enzyme exhibits an increasing $H_2$-oxidation rate with increasing buffer pH[6]. Accordingly, the pH range used in this assay was extended to pH 10. As shown in Supplementary Fig. 3, $H_2$-uptake activity of native HydA1 generally enhanced with decreasing $H_3O^+$ concentrations, exhibiting a nearly fivefold rate-increase per pH unit between pH 6 and 8. After an intermediate drop between pH 8 and 9 the $H_2$-oxidation activity further increased to nearly 20.000 μmol $H_2$ per mg per min between pH 9 and 10. The absolute activities of SDM variants were significantly diminished compared to wild-type HydA1, reaching a pH optimum of at best 6% (R148A). Up to pH 9, variant R148A$_{HydA1}$ showed a wild-type-like trend for the pH-activity profile of relative $H_2$-oxidation activity, while instead of a second increase between pH 9 and 10, the activity strongly declined, suggesting that this second increase is connected with the titration of the guanidine base of R148. The pH-activity profile of E144A$_{HydA1}$ was quite similar to wild-type HydA1, despite a flattening out of the local maximum at around pH 8. The latter suggests that this local activity maximum is depending on the presence of the surface-exposed Glu residue. S189A and C169D only showed single activity peaks at pH 8 or 9, respectively with very low $H_2$-uptake rates of 1–2%, compared to wild type.

**Infrared spectroscopy**. We recently showed that decreased proton release efficiency in the presence of $H_2$ leads to an accumulation of the hydride state, $H_{hyd}$[29]. This H-cluster intermediate carries a terminal hydride and represents the first catalytic intermediate after heterolytic cleavage of $H_2$[26–29]. In previous studies we exploited this behavior to demonstrate that positions C169$_{HydA1}$/C299$_{CpI}$ and E141$_{HydA1}$/E279$_{CpI}$ contribute to catalytic PT[29]. Non-conservative substitutions to alanine that interrupted the PT pathway led to an enrichment of $H_{hyd}$ instead of adopting a mix of reduced redox species (see below). Here this approach was applied to link PT activity of residues in the putative PT pathway to the steady-state equilibrium of redox species in the presence of $H_2$. Employing ATR-FTIR spectroscopy, we probed CpI and HydA1 wild-type protein and the respective SDM variants at pH 8 under either $N_2$ or $H_2$ (Fig. 3). When flushed with $N_2$, auto-oxidized HydA1 wild type and SDM variants uniformly exhibit peaks around 1964, 1939, and 1802 cm$^{-1}$ characteristic for the $H_{ox}$ state[34]. HydA1 and CpI wild-type enzymes adopt a mixture of reduced states when exposed to $H_2$, predominantly $H_{red}$/$H_{sred}$ and $H_{red}'$[34,35]. In contrast, most variants populate the $H_{hyd}$ state that can be enriched in wild-type enzyme only at pH 4 (see Supplementary Fig. 4). Similar results were achieved for the corresponding variants of CpI as shown in Fig. 3c, d. HydA1 variants C169D, S189A, E144D, and R148A, as well as corresponding CpI variants show wild-type-like spectra by adopting a mixture of reduced states instead of $H_{hyd}$ (Fig. 3b). With the exception of S189A$_{HydA1}$/S319A$_{CpI}$, these variants retain a higher level of $H_2$-release activity (Fig. 1b), which explains why in these cases $H_{hyd}$ cannot be trapped at pH 8; just as wild-type enzyme, these variants populate $H_{hyd}$ when titrated to significantly lower pH values (Supplementary Fig. 4). Interestingly, variant C169D and the corresponding variant of CpI (C299D) seem to be incapable of accumulating $H_{hyd}$ under any of the conditions applied here.

**Protein crystallography**. For eight CpI variants (holoenzyme) and three HydA1 variants (apo-enzyme) protein crystals were obtained. Their structures were solved and refined to resolutions of 1.45–2.76 Å, allowing us to gain insight into the structural consequences of SDM. Corresponding to earlier crystal structure data, for all CpI variants a space group of P1 $2_1$ 1 was observed with two copies in the asymmetric unit cell (chain A and B)[19,25].

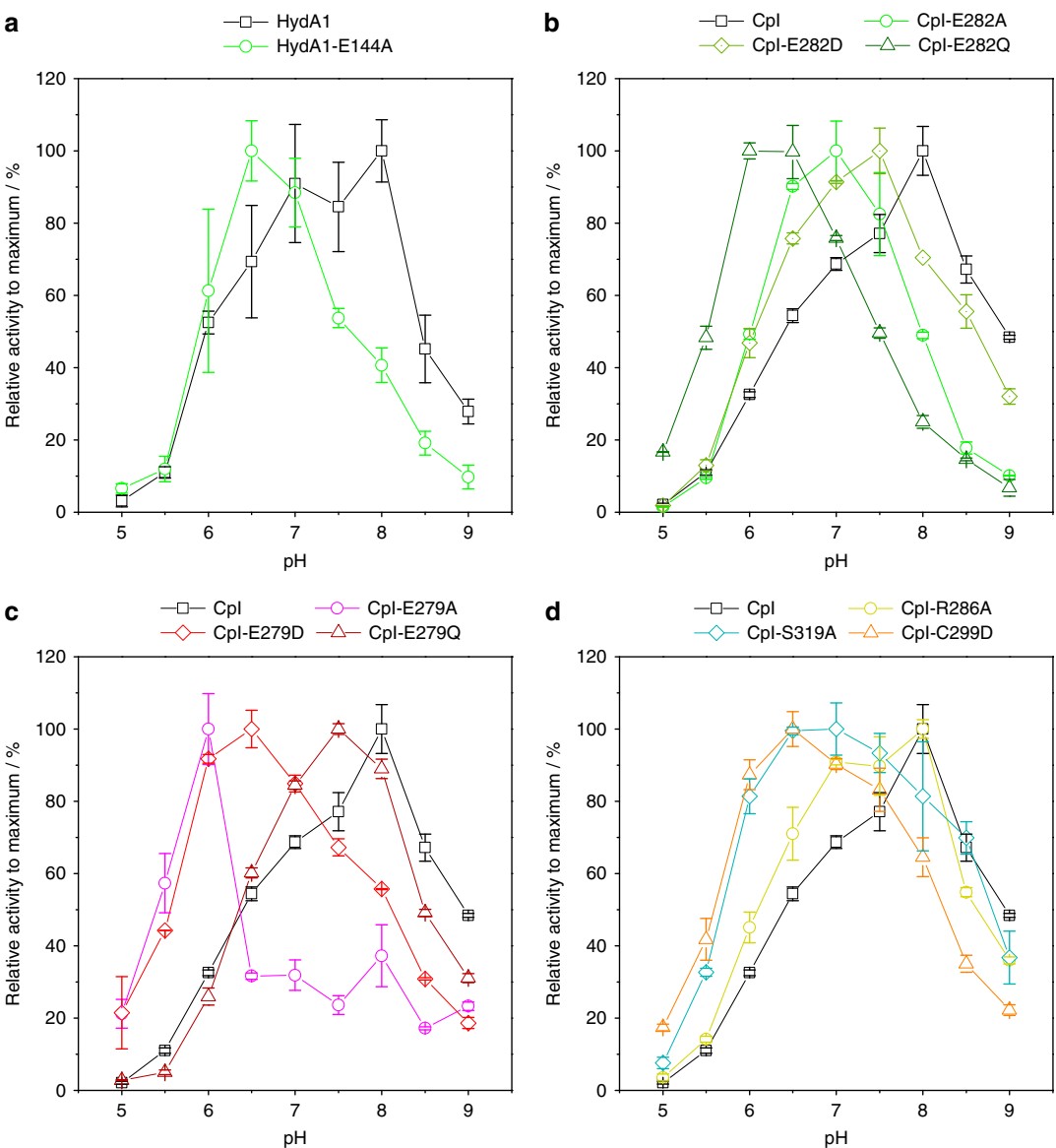

**Fig. 2** pH-activity profiles of wt-CpI and wt-HydA1 compared to selected SDM variants. **a** HydA1 and E144A; **b** wt-CpI and variants of position E282; **c** wt-CpI and variants of position E279; **d** wt-CpI and selected variants of positions R286, S319, and C299. $H_2$ production activities were determined for different buffers, covering a pH gradient between 5 and 9 in steps of 0.5 pH units. Relative values correspond to % of maximum activity obtained throughout the entire pH gradient. Black plots indicate the relative pH-dependent activities of CpI and HydA1 wild-type enzymes. All values are mean values ± standard deviations from at least three independent measurements

Crystallization of HydA1 exclusively succeeded for apo-protein, which carries the $[4Fe]_H$-cluster but lacks the $[2Fe]_H$-site (Supplementary Fig. 5). However, comparisons of crystal structures of CpI apo- and holo-protein with apo-HydA1 demonstrated that a lack of cofactor does not affect the configuration of the putative PT pathway[19,36]. Variants of HydA1 apo-protein were crystallized in space group P3$_2$ 2 1, with a single chain in the asymmetric unit. Overall, SDM did not induce unspecific structural changes as deduced from superposition with wild-type enzyme and corresponding root-mean square deviations of Cα atoms (Supplementary Table 2) and the the H-cluster was largely present for all CpI variants (Supplementary Table 5).

In the following, local structural changes in the putative PT pathway will be described. We start from the surface-exposed residues R286$_{CpI}$ and E282$_{CpI}$, will continue addressing the median positions S319$_{CpI}$ and E279$_{CpI}$, and end with C299$_{CpI}$ in the vicinity of the H-cluster.

According to their close inter-residue distance of 2.8 Å, R286$_{CpI}$ may function as a salt bridge partner of deprotonated E282$_{CpI}$ (Fig. 4, right and Supplementary Figs. 6, 7) and thus could be involved in the PT mechanism. R286$_{CpI}$ is further part of an extensive hydrogen-bonding network, which includes the carboxyl group of E282$_{CpI}$, histidine residues H565$_{CpI}$ and H569$_{CpI}$, and surface-bound $H_2O$ molecules. In case of variant R286A significantly fewer water molecules are observed in the same region (Supplementary Fig. 8). We therefore assume that the loss of the guanidine group in R286A destabilizes the network of proton-accepting/donating surface-bound $H_2O$ molecules near the entrance to the PT pathway. However, as the essential chain of PT pathway residues is not directly affected, the influence of this exchange on catalytic activity is comparatively weak (Fig. 1b).

In E282A, two water molecules occupy the space of the missing carboxylic acid group (chain B, Wat717 and Wat974) (Fig. 4, top middle and Supplementary Fig. 13a). The distances between these

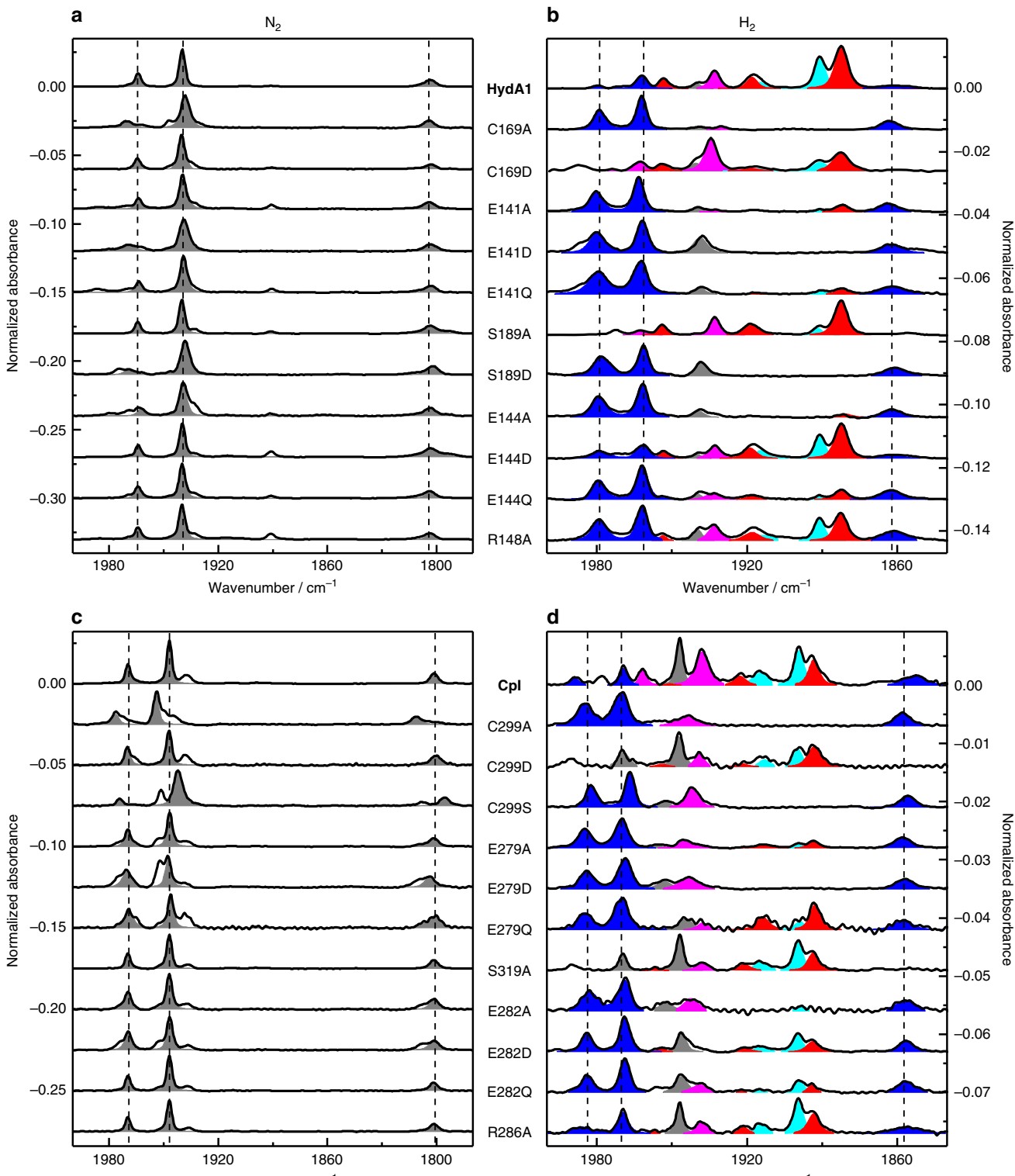

**Fig. 3** Infrared spectra of wild types and PT pathway variants. The frequency regime of the H-clusters' CO ligands is shown (2000–1785 cm$^{-1}$). For ATR-FTIR spectroscopy, the buffer was set to pH 8 and the rehydrated samples were purged with 100% H$_2$ for 5 min. **a**, **c** Auto-oxidation in absence of H$_2$ (i.e. purged with N$_2$) was exploited to likewise enrich for all examined proteins the oxidized resting state, H$_{ox}$ (gray bands). Some CpI variants tend to accumulate H$_{ox}$H in parallel with H$_{ox}$ (e.g. C299A bands at frequencies 1975/1953/1809)[34]. **b**, **d** When shifting from N$_2$ to H$_2$ the spectrum of wild-type protein changes to different fractions of reduced species including H$_{red}$ (cyan) and H$_{sred}$ (red) as well as H$_{red}$´ (magenta). Most PT pathway variants accumulate H$_{hyd}$ (blue) instead or in addition to a mix of reduced states. For precise state-specific vibrational signals of CpI and HydA1 see Supplementary Table 7

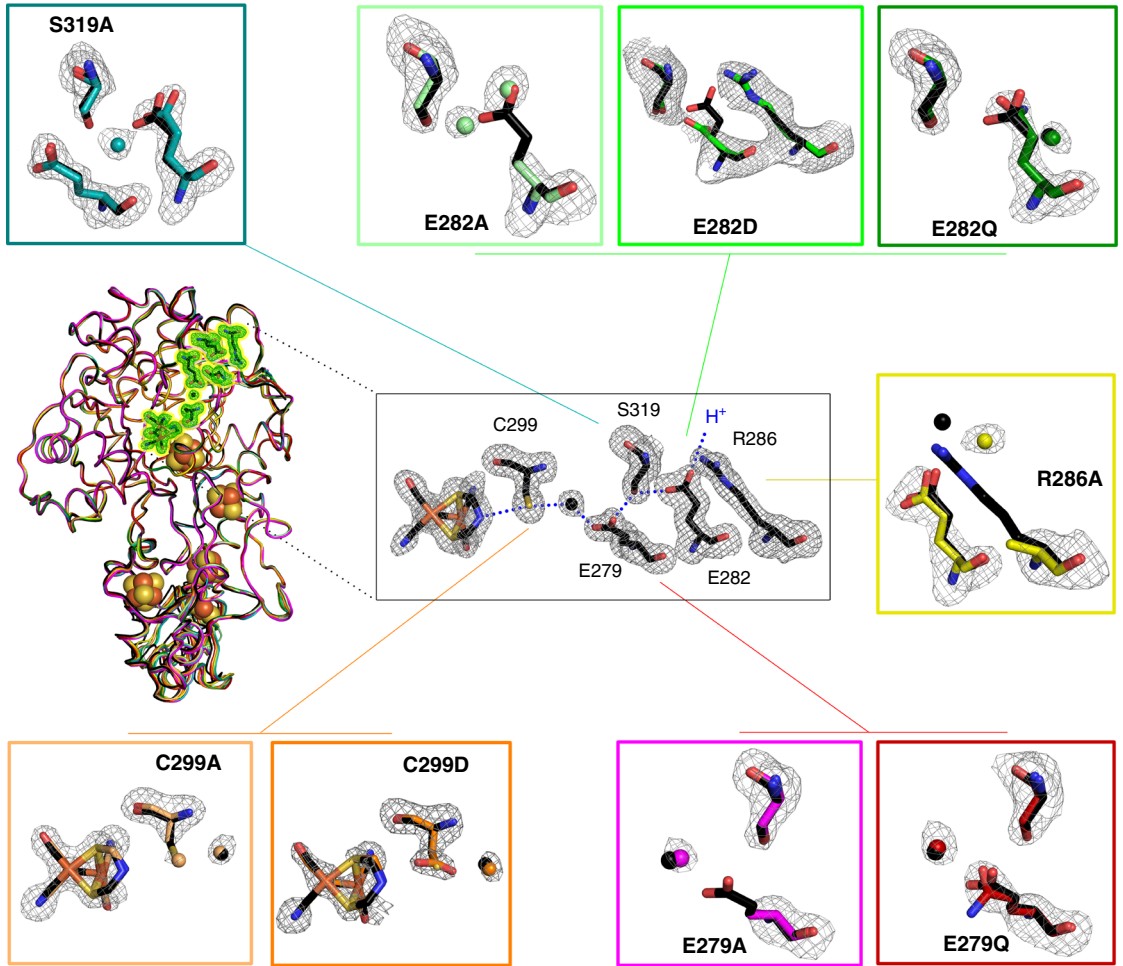

**Fig. 4** Structural features of SDM variants targeting the putative PT pathway in CpI. Structures of nine SDM variants are superimposed as cartoon-loop models together with the 4XDC[19] wild-type structure. No unspecific differences are observed. For each variant an enlargement of its electron density map in the putative PT pathway and the corresponding sticks model has been aligned with the structure of wild-type protein ($H_2O$ molecules and carbon atoms colored in black). For the CpI proteins local structural differences near the site of mutagenesis are depicted (for the HydA1 variants see Supplementary Fig. 5). Simulated annealing omitting maps ($F_o - F_c$) were contoured at $3\sigma$ except for E282D, which was contoured at $1.9\sigma$ due to its comparatively low resolution. Chain B provides a more flexible N terminus but a more rigid H-domain where both the PT pathway and the active center are located[19]. Therefore, all the structural information of CpI was derived from chain B if not stated otherwise

water molecules and the hydroxyl group of S319 are 2.5 and 3.1 Å (see Supplementary Fig. 7), respectively. They are close enough to rescue the PT activity between surface water and $S319_{CpI}$ further downstream the PT pathway. In the crystal structure of apo-$E144A_{HydA1}$, one water molecule remains in a very similar position (Supplementary Fig. 5). It is therefore not surprising that both variants exhibit 50% wild-type activity (Fig. 1). In variant $E282Q_{CpI}$, the glutamine residue is potentially stabilized by two hydrogen bonds (H-bonds) unrelated to the putative PT pathway thus, blocking PT and rendering the enzyme largely inactive (Supplementary Fig. 7). Additionally, the two different conformations of $Q282_{CpI}$ in chain A and B (Supplementary Fig. 7) indicate structural flexibility at the entrance of the PT pathway, which may support the residual activity measured for this variant.

In variant $S319A_{CpI}$, the carboxyl group of $E282_{CpI}$ is slightly shifted outward, probably due to the lack of the hydroxyl group at position $319_{CpI}$. In wild-type enzyme $S319_{CpI}$ acts as H-bond donor to $E282_{CpI}$ and drags its carboxy group further into the PT pathway (Fig. 4). In $S319A_{CpI}$ an unprecedented water molecule (Wat735) is located between E282 and A319, and the nearby water molecule Wat873 of wild-type CpI (chain B) is missing here, suggesting a translocation enabled by the unoccupied space

of the missing hydroxyl group of position 319 and the slight outward shift of E282 (see Supplementary Fig. 12c). Although the distance to the carboxyl group of E279 amounts to 5.8 Å, the presence of Wat735 may explain the dramatically diminished yet still detectable $H_2$ release activity of variant $S319A_{CpI}$ (6%).

The conformational differences between $E279Q_{CpI}$ and wild-type CpI are insignificant, except that the carboxamide group of $Q279_{CpI}$ is slightly twisted relative to the original carboxyl group (Fig. 4). In the corresponding HydA1-variant E141Q, the glutamine residue precisely adopts the conformation of the glutamic acid residue in HydA1 wild type (Supplementary Fig. 5). For the corresponding alanine variant of both, HydA1 and CpI, shifts of the conserved water molecule in the putative PT pathway can be observed. While glutamine occupies the entire space of the native carboxyl group and thus clearly interrupts the putative PT pathway, an exchange to alanine may support additional $H_2O$ molecules to bridge the gap. However, in contrast to corresponding substitutions at the surface-exposed glutamate $E282_{CpI}$, in neither $E141A_{HydA1}$ (Supplementary Fig. 5) nor $E279A_{CpI}$[25] (Fig. 4) additional $H_2O$ molecules are observed. This explains why exchanges of the median glutamate to either alanine or glutamine lead to residual activities of below 1% (Fig. 1).

In case of C299D$_{CpI}$, the carboxyl group of the side chain establishes H-bonds (2.7 Å) with the amine group of [2Fe]$_H$ and the "conserved" water molecule Wat708 (corresponding to Wat826 in chain B of 4XDC[19]) (Fig. 4). Unsurprisingly, protein samples of C169D$_{HydA1}$ and C299D$_{CpI}$ both retain a rather high level of H$_2$ release activity (30–80%, Fig. 1). This is not the case when substituting cysteine for alanine, which for both enzymes leads to a complete loss of catalytic activity. Interestingly, the crystal structure of C299A$_{CpI}$ reveals that the removal of the thiol group causes an additional H$_2$O molecule to occupy the vacant space. While the additional H$_2$O is expected to rescue PT activity, it obviously does not restore enzymatic activity. A summary of the characteristic experimental data gained for each enzyme variant is presented in Supplementary Table 1.

## Discussion

Fast proton transfer between bulk solvent and the H-cluster is a precondition for the high catalytic turnover rates of [FeFe]-hydrogenases. Mainly based on theoretical studies, four potential PT pathways have been discussed in literature[15–18,20,37,38]. The putative trajectory examined in this study has been favored as key catalytic PT pathway. However, unique experimental evidence for this assumption was lacking. A characterization of PT does not only help to understand the catalytic mechanism but will further contribute valuable parameters for the design of bio-inspired inorganic catalysts[20]. For artificial hydrogen catalysts, proton supply is often rate-limiting, e.g. due to the lack of a defined proton relay, thus making it necessary to add strong acids and adjust a very low solvent pH[39]. In this study, corresponding sets of SDM variants of CpI and HydA1 were generated to uncover the individual contribution of residues to PT in the putative PT pathway of [FeFe]-hydrogenases[25,40]. The complementary data gained from pH-activity profiles, crystal structure analysis, and ATR-FTIR spectroscopy unanimously show that the residues of this pathway are key for the supply and release of protons during catalytic turnover.

The crystal structures of wild-type CpI and its SDM variants reflect the H$_{ox}$ state[19], which is commonly accepted to be the resting state of hydrogen turnover. To obtain a general survey on the putative hydrogen-bonding pattern in H$_{ox}$, the distances between consecutive PT partners in wild-type CpI and the nine SDM variants were derived from the corresponding structures (Fig. 5 and Supplementary Fig. 7). On the basis of CpI crystal structure 4XDC[19], PROPKA empirically predicted $p$Ka values of 8.6 and 3.5 for E279 and E282, respectively (pH 8)[41]. The relatively high $p$Ka of E279$_{CpI}$ reflects its hydrophobic environment

compared to the surface-exposed E282$_{CpI}$ and suggests E279$_{CpI}$ to be largely protonated while E282$_{CpI}$ probably resides in the deprotonated state. Starting at the H-cluster, possible hydrogen-bonding interactions can be assumed for adt-NH/C299$_{CpI}$ and C299$_{CpI}$/Wat826, both of which exhibit comparatively large distances of 3.5 and 3.2 Å[42]. Molecular dynamics simulations proposed a low H-bond occupancy between adt-NH and the thiol group of C299$_{CpI}$ and a stronger contact between E282$_{CpI}$ and R286$_{CpI}$ to be preconditions for H$_2$ oxidation starting from H$_{ox}$[20]. This proposal cannot be confirmed by our crystallographic data, which show the C299 thiol group in an intermediate orientation. However, we cannot rule out the possibility that such differences between theoretical data and structural information are influenced by the non-physiological crystallization conditions or low temperature crystal storage in liquid nitrogen, which might favor certain configurations in the PT pathway. The short distance of 2.5 Å between Wat826 and E279$_{CpI}$ suggests a strong H-bond, in contrast to the adjoined pair E279$_{CpI}$/S319$_{CpI}$, for which a distance of 3.6 Å again indicates a fairly weak interaction. The putative hydrogen bond between S319/E282 can be estimated to be of moderate strength (2.9 Å). E282$_{CpI}$ potentially interacts with at least one surface-bound H$_2$O molecule (Wat990). Finally, arginine R286$_{CpI}$ (calculated $p$Ka: 12.4) may serve as a putative salt bridge or hydrogen-bonding partner at close distance to E282$_{CpI}$ (2.8 Å). The combination of calculated $p$K$_a$ values and H-bond distances described in this study suggests for H$_{ox}$ the H-bond pattern depicted in Fig. 5.

Electron densities in the omit maps of crystal structures for wild-type CpI and most of the SDM variants examined here are unambiguously oriented (Fig. 6) and show low displacement factors (Supplementary Table 8). This is in contrast to quantum mechanics/molecular mechanics simulations by Long et al.[38] who proposed a conformational bi-stability for the two glutamic acid residues E282 and E279 in the catalytic PT pathway. Our data rather favor a model in accordance with the Grotthuss mechanism, based on simultaneous deprotonation/protonation events according to a bucket line between strictly orientated residues and water molecules[43]. However, a bi-stability of the glutamate residues under turnover conditions might still be a valid interpretation, as it could also be regarded as switching between two different hydrogen-bonding patterns. In both cases, the direction of PT would be solely determined by the redox state of the [2Fe]$_H$-cluster (see Supplementary Fig. 9 and Supplementary Discussion).

The FTIR data presented here were recorded under steady-state conditions applying changes in gas atmosphere and pH. As protons are reactants in hydrogen turnover (H$_2 \rightleftharpoons$ 2H$^+$ + 2e$^-$), it

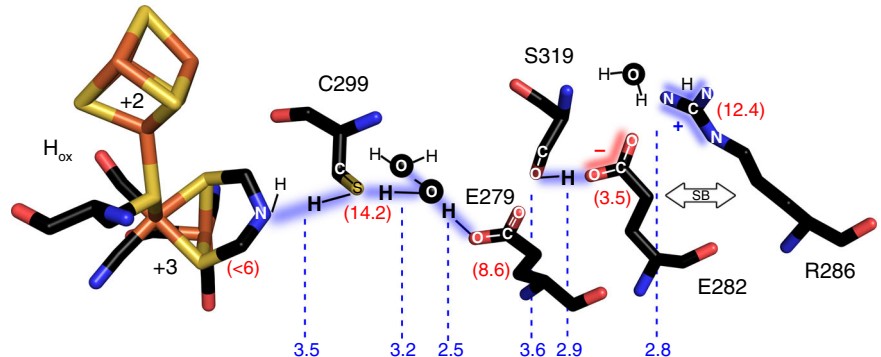

**Fig. 5** Hydrogen-bonding pattern in the catalytic PT pathway of the H$_{ox}$ state. H-bond pattern of H$_{ox}$ according to $p$Ka values of the residues calculated for the structure of wild-type CpI (4XDC[19]) via PROPKA[41] (see red numbers in parentheses). The $p$Ka of the adt-ligand in H$_{ox}$ is derived from a previous study[58]. The blue numbers indicate the distances between neighboring positions of the PT pathway. The arrow labeled "SB" indicates a presumptive salt bridge contact between R286 and deprotonated E282

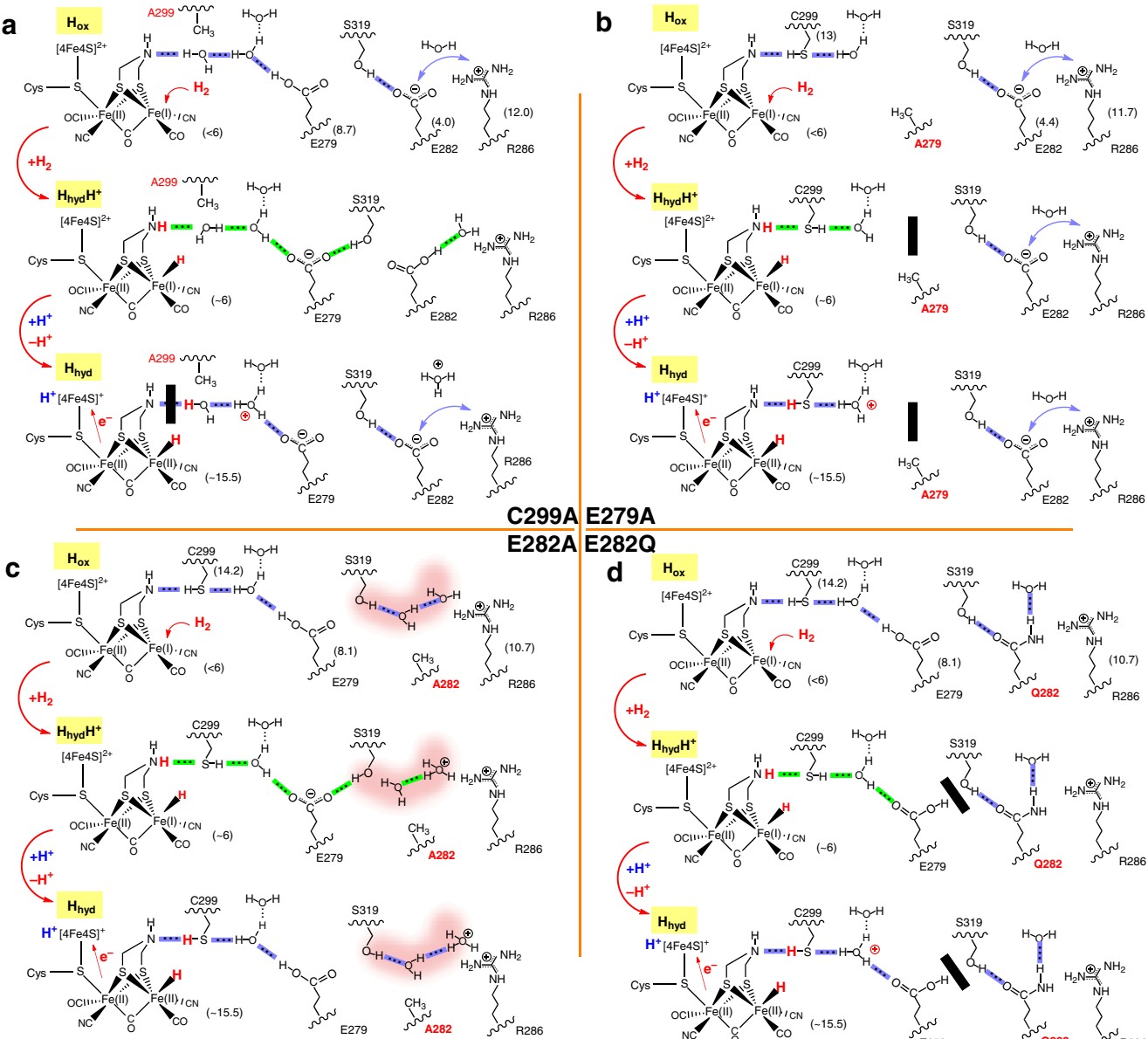

**Fig. 6** Influence of SDM on the proton transfer mechanism during $H_2$-uptake in CpI. Effects of SDM on the proposed proton transfer mechanism for selected CpI variants: C299A (**a**), E279A (**b**), E282A (**c**), and E282Q (**d**). $H_2$-binding induces a shift in the H-bond pattern (from mode 1 to mode 2) and initiates the catalytic mechanism during which the H-bond pattern repeatedly shifts between modes 1 (blue) and 2 (green) while promoting a stepwise proton release via the PT pathway (for details see Supplementary Fig. 9+14). The pKa values of adt-ligand at different redox states are derived from previous studies[49,58]. The mutated residues and hydrogen atoms from substrate were highlighted as red. The green double arrow indicates a putative salt bridge contact. In **c** (E282A), the pink shading area indicates a slowed-down but still functioning proton transfer. Protons presented in blue close to the $[4Fe]_H$ sub-cluster originate from the recently described regulatory PT pathway[34,59,60], which is independent of substrate/product transfer

is plausible to assume that mutagenesis in the catalytic PT pathway influences the equilibrium of H-cluster species. This was exemplified above for those variants that accumulate $H_{hyd}$. A similar effect on the dynamic equilibrium of catalytic states has been observed for [NiFe]-hydrogenases[44,45]. The substitution of E17 for glutamine in the putative PT pathway of soluble hydrogenase I (SHI) of *Pyrococcus furiosus* disabled proton-coupled electron transfer (PCET) between two catalytic states ($Ni_a$-C and $Ni_a$-S)[44], which was indicative by an accumulation of intermediates $Ni_a$-$I^1$ and $Ni_a$-$I^2$. For [FeFe]-hydrogenases, the hydride state has been demonstrated to accumulate under $H_2$-oxidation conditions if proton efflux is severely restricted. This may be the result of an oversaturation of the native PT pathway due to the

enhanced proton pressure, i.e. low bulk pH[29,34]. It can further be the result of $[2Fe]_H$ derivatization or eliminating protonatable side chains in the catalytic PT pathway[29]. The hydride state has been shown to exhibit an uncharged azadithiolate ligand (adt-NH)[37,46] that implies a transient intermediate with a protonated ammonium cation, e.g. as proposed by Reijerse et al. ($H_{hyd}H^+$)[27]. Accordingly, oxidation of $H_{hyd}$ can be explained by PCET that relies on a functional release of protons. All PT pathway variants specifically react with $H_2$ and it can be concluded that at least one proton is always injected into the hydrogen-bonding network, despite the compromised PT pathway. Single deprotonation appears possible as the next proton-binding site downstream $C299_{CpI}$ would be Wat826/Wat1120, which could form a Zundel-

ion-like configuration upon protonation[47] ($H_5O_2^+$, Fig. 5 and Supplementary Fig. 12a). Neither of the two $H_2O$ molecules has been directly affected by mutagenesis. It may be concluded that PT is only blocked if a protonation of Wat826/Wat1120 yield a Zundel-ion-like configuration, stabilized due to increased $pK_a$ in comparison to $H_3O^+$.

The favorable position of the acidic asparagine side chain in the structure of $C299D_{CpI}$ effectively connects adt-NH of the $[2Fe]_H$ moiety with the conserved water molecule Wat708 in the otherwise intact PT pathway, rendering this variant significantly active. This comparatively high level of absolute activity and the fact that both corresponding variants, $C169D_{HydA1}$ and $C299D_{CpI}$, exhibit significant shifts in their pH-dependent activity optimum to the acidic range might explain why this type of variant refuses to accumulate $H_{hyd}$ even at pH 4, while instead effectively continuing the turnover process as evident according to the comparatively dominant infrared bands for the reduced states $H_{red}$ and $H_{sred}$ (1890 and 1882 cm$^{-1}$ for C169D) recorded at pH 4 (see Supplementary Fig. 4). In case of variant $C299A_{CpI}$, the first deprotonation step ($H_{hyd}H^+ \rightleftharpoons H_{hyd} + H^+$) seems impossible but Fig. 3 shows that this variant very effectively accumulates $H_{hyd}$ under $H_2$. Thus, an alternative proton acceptor has to be assumed. The additional $H_2O$ molecule (Wat962), which is trapped in the space of the missing thiol group of variant C299A would be a plausible proton acceptor (Fig. 6 and Supplementary Fig. 13b). As it is well-positioned to bridge the adt-ligand (distance 3.4–3.7 Å) and Wat826 (distance 3.6–3.7 Å), it seems surprising that we are unable to measure any significant catalytic activity. This could be due to an unfavorable $pK_a$ difference among the Zundel-ion-like complex, the uncharged adt-ligand and glutamic acid $E279_{Cp}$[48]. The $pK_a$ calculated for the adt-ligand in $H_{hyd}$ is significantly larger than the one determined for $H_{ox}$ in mimics of the $[2Fe]_H$-cluster[49]. However, no re-protonation of $H_{hyd}$ is observed. We assume that the proton is trapped in a Zundel-ion-like configuration with a dangling $H_2O$ molecule Wat962 downstream of position A299 that prevents E279 from being re-protonated (Fig. 6a, C299A). Upon oxidation of $H_2$, proton release via the H-cluster may restrict re-protonation of $H_{hyd}$ and steady-state accumulation of $H_{hyd}H^+$. Recently, an alanine was identified at the position homologous to $C299_{CpI}$ in the newly described sensory [FeFe]-hydrogenase HydS of Thermotoga maritima, which showed very low $H_2$ release and oxidation activities (<5% of HydA1)[50]. However, HydS is clearly more active than the C $\rightarrow$ A variants of CpI and HydA1 suggesting a slightly different situation for HydS. Nevertheless, the very low activity level of this sensory-type [FeFe]-hydrogenase overall agrees very well with our results.

For variant $E279A_{CpI}$ the deprotonation of the $[2Fe]_H$-cluster, which produces the hydride state would lead to the formation of the Zundel-ion-like configuration. However, due to the large distance between Wat875 and S319 (up to 4.5 Å) the proton cannot proceed any further. As a consequence, $H_{hyd}$ with its deprotonated adt-NH ligand would be stabilized in presence of $H_2$ (Fig. 6b, E279A). In case of the corresponding exchange at the surface-exposed glutamate in $E282A_{CpI}$, two $H_2O$ molecules (chain B: Wat717 and Wat974) invading from the solvent are able to bridge the gap of the missing carboxyl group (Fig. 6c, E282A and Supplementary Fig. 13a) thereby rescuing a large fraction of $H_2$-release activity (~50%). The fact that $E282A_{CpI}$ is rescued by an $H_2O$ moiety while in $E279A_{CpI}$ and $C299A_{CpI}$ the gap in the catalytic PT pathway cannot be closed, is probably connected to the superior accessibility at the protein surface and the specific functional requirements (such as e.g. $pK_a$) of the position to be rescued within the PT pathway chain. This illustrates the significance of individual residues for the catalytic PT pathway, which appears to increase from surface to catalytic center.

However, the lack of $H_2$ release activity resulting from the non-conservative exchange in SDM variant $E282Q_{CpI}$ (Fig. 6d, E282Q), which prohibits $H_2O$ access due to steric reasons demonstrates the necessity of a protonatable position at the entrance of the catalytic PT pathway (for SDM variants E279Q and S319A see Supplementary Figs. 10-11). Supplementary Table 9 provides a survey of the presumptive functions of all positions in the PT pathway, according to the conclusions drawn here.

The multilayered approach followed here, comprising enzyme kinetics, ATR-FTIR spectroscopy, and crystal-structure analysis, unambiguously verifies the herein proposed PT pathway to be the main trajectory for substrate/product transfer in [FeFe]-hydrogenases. From surface to active center the impact of substitutions along the PT pathway increases, with the inner part between $E279_{CpI}$ and $C299_{CpI}$ being the most vital positions for basic PT function. A model of the PT mechanism in line with the presented data further suggests a major role for the $H_2O$-cluster and a Zundel-ion-like configuration comprising Wat826 and Wat1120 in regulating PT pathway function. External $H_2O$ molecules can rescue PT function (1) if provided sufficient space and accessibility, (2) if gaps to neighboring PT pathway positions do not exceed H-bond distance ($\leq 4$ Å), and (3) if the overall $pK_a$-gradient of the PT chain is not severely imbalanced. How PT is precisely coupled and synchronized with electron transfer between $[2Fe]_H$ and $[4Fe]_H$ remains to be further elucidated, possibly by a time-resolved characterization of the catalytic mechanism under sub-turnover conditions. The variants described here are dramatically slowed down in PT efficiency and provide excellent models to resolve the individual steps. In conclusion, the herein presented data give valuable insights into the molecular parameters that enable and tune PT in redox active proteins, thus providing useful guidelines for de novo catalyst design.

## Methods

**Site-directed mutagenesis.** Expression constructs were generated using the QuikChange® Site-Directed Mutagenesis Kit from Stratagene using either pET21b-HydA1Cr or pET21b-CpI as template and corresponding mismatch primers[29], listed in Supplementary Table 6. The resulting constructs for the expression of SDM variants were verified via sequencing (3130xl Genetic Analyzer; Applied Biosystems) and used for the transformation of Escherichia coli expression host strain BL21 (DE3) $\Delta iscR$[51], which was kindly provided by Patrik R. Jones.

**Protein expression and maturation.** Handling of expression cultures and enzyme variants was entirely done under strictly anaerobic conditions in a glove box (Coy laboratory) atmosphere of 98.5% $N_2$ and 1.5% $H_2$. HydA1 and CpI proteins were expressed in absence of maturases HydE, -F, and -G[52] as described previously and therefore lack the catalytically essential $[2Fe]_H$ sub-cluster in the original state after purification. Cell disruption was achieved by ultrasonication. Cell debris was separated from the supernatant by ultracentrifugation and filtration (pore size 0.2 µm). Protein purification was done employing strep-tactin/strep-tag II (IBA GmbH) affinity chromatography. HydA1 and CpI apo-proteins were matured in vitro by adding the synthetic mimic of the native $[2Fe]_H$-complex ($Fe_2[\mu-(SCH_2)_2NH]$ $(CN)_2(CO)_4[Et_4N]_2)$ ($[2Fe_H]^{MIM}$) at a 10-fold molar excess[31]. $[2Fe_H]^{MIM}$ was synthesized following literature procedures and kept in 100 mM $K_2HPO_4/KH_2PO_4$ buffer (pH 6.8) at $-80$ °C[53]. After an incubation period of 1 h at 25 °C, size-exclusion chromatography was employed to remove redundant complex, using NAP-5 columns (GE Healthcare), which were equilibrated with 100 mM Tris–HCl (pH 8) containing 2 mM sodium dithiolate (NaDT) prior to use. Proteins were concentrated using 30 kDa Amicon Ultra centrifugal Filter units (Merck Millipore) and stored anaerobically at $-80$ °C. SDS-polyacrylamide gel electrophoresis and Bradford assays were employed to verify protein purity and to calculate the resulting protein concentration.

**Activity assays.** $H_2$-evolution rates of enzyme variants (0.4–4 µg, depending on the level of residual activity) were determined employing a standard in vitro assay, comprising 100 mM NaDT as sacrificial electron donor and 10 mM methyl viologen as electron mediator in 0.1 M $K_2HPO_4/KH_2PO_4$, pH 6.8. The reaction volume of 2 ml was prepared in air-tight suba-seal vessels and purged with argon for 5 min prior to the $H_2$-production period of 20 min in a shaking water bath, kept at 37 °C.

The $H_2$-production yield was determined by analyzing 400 μl of the sample head-space via gas-chromatography (GC-2010, Shimadzu). For the determination of pH-activity profiles, four different buffers were used and adjusted to individual pH values according to their respective buffer range, covering pH 5–10 with a resolution of 0.5 pH units: pH 5–6.5 (200 mM MES-NaOH, 2-(N-morpholino) ethanesulfonic acid); pH 7–7.5 (200 mM MOPS-NaOH, 3-(N-morpholino) propanesulfonic acid); pH 8–9 (200 mM Tris-HCl); and pH 10 (200 mM CAPS-NaOH, N-cyclohexyl-3-aminopropanesulfonic acid). pH-dependent $H_2$-uptake activity was monitored performing a colorimetric assay at 25 °C with enzyme amounts between 0.5 ng and 1 μg in an atmosphere of 1 bar $H_2$, using 10 mM benzyl viologen (Sigma-Aldrich) as electron acceptor ($\varepsilon_{600}$: 10 mM$^{-1}$ cm$^{-1}$)[6].

**Crystallization and structural determination of variants**. Vapor diffusion hanging drop and sitting drop methods were applied under anaerobic conditions at 277 K to crystallize the SDM variants using protein concentrations between 10 and 15 mg ml$^{-1}$. Details of the crystallization conditions are summarized in Supplementary Table 3. Mounting was carried out after 5–10 days of crystal growth and crystals selected for data collection were flash-frozen and stored in liquid $N_2$. Diffraction data were collected at 100 K in different beamlines as indicated in Supplementary Table 3 and processed using XDS[54]. Phenix[55] and Coot[56] were employed for molecular replacement (starting models for CpI and HydA1 were 4XDC[19] and 3LX4[36] respectively) and structural refinement. The details of crystallographic data gained for each of the variants are summarized in Supplementary Table 4.

**Infrared spectroscopy**. Wild-type and SDM variants of HydA1 and CpI were probed by in situ ATR-FTIR spectroscopy[29,34,57] using a rapid-scan Tensor 27 spectrometer (Bruker Optik, Germany) equipped with a three-reflection ZnSe/silicon crystal ATR cell (Smith Detection, USA). The spectrometer was kept under anaerobic conditions in a vinyl glove box (Coy Laboratories, USA) under a water-free atmosphere of 99% $N_2$ and 1% $H_2$. All experiments were performed at room temperature, on hydrated films and in the dark. The oxidized state $H_{ox}$ was populated under 100% $N_2$ ambient partial pressure while the reduced states ($H_{red}$/$H_{sred}$, $H_{red}'$, and $H_{hyd}$) were observed in the presence of $H_2$ exclusively. All spectra shown in Fig. 3 were mathematically corrected for the broad combination band of liquid $H_2O$ at around 2120 cm$^{-1}$ and normalized to unity.

**Reporting Summary**. Further information on research design is available in the Nature Research Reporting Summary linked to this article.

## Data availability

The coordinates and structure factors for all structures are deposited in the PDB as 6GLY, 6GLZ, 6GM0, 6GM1, 6GM2, 6GM3, 6GM4, 6GM5, 6GM6, 6GM7, and 6GM8. Further data supporting findings of this study are available from the corresponding authors upon reasonable request. A Reporting Summary for this Article is available as a Supplementary Information file.

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

## Acknowledgements

M.W. and T.H. gratefully acknowledge financial support from the Volkswagen-Stiftung (Design of [FeS]-cluster containing Metallo-DNAzymes (Az 93412)) Deutsche Forschungsgemeinschaft (DFG) (the Cluster of Excellence RESOLV EXC1069 and GRK 2341: Microbial Substrate Conversion). J.D. acknowledges support by the China Scholarship Council (CSC) and DFG-EXC1069. M.S. and S.T.S. thank the International Max Plank Research School (IMPRS) on Multiscale Biosystems and the Focus Area NanoScale (Freie Universität Berlin) for financial support. U.-P.A. and F.W. are grateful for financial support by the Deutsche Forschungsgemeinschaft (Emmy Noether grant to U.-P.A., AP242/2-1) and the Fraunhofer Internal Programs (Attract 097-602175). F.W. further thanks the Studienstiftung des deutschen Volkes for a PhD fellowship. We thank the staff of beamlines ID23-1 and BM30A at the ESRF and PXII at the SLS for technical support during X-ray data collection.

## Author contributions

M.W. and T.H. designed the research. M.W., J.D., S.T.S., and M.S. wrote the manuscript. J.D. expressed, purified, and maturated the proteins and biochemically characterized them. J.D., J.E., V.E., and E.H. performed protein crystallization and structural analysis. M.S. and S.T.S. were responsible for infrared spectroscopy part. F.W. and U.-P.A. synthesized the $[2Fe]_H$ complex.

## Additional information

**Competing interests:** The authors declare no competing interests.

