## [Peer Review File · Nature Communications]

Reviewers' comments:

Reviewer #1 (Remarks to the Author):

Review for Crystallographic and spectroscopic assignment of the proton transfer pathway in [FeFe]-hydrogenases

Content:

The powerful tool of site directed mutagenesis was used to investigate the PT-pathway of Cpl and HydA1. This resulted in 11 mutants for two [FeFe]-H₂ases, characterized by FTIR and 11 crystal structure analyses as a basis for interpretation of enzyme kinetics in H₂ production and oxidation/proton release activity. The overall review of the design of the diiron hydrogenase enzyme, proton delivery or removal route, is put on very firm ground. In addition it verifies the convergent evolution not only of the active sites, but also of the outer machinery that makes the enzymes so efficient for hydrogen processing.

Overall Evaluation:

Novelty : medium to high (the idea of site-directed mutagenesis of amino acids in PT-pathway is not new; several studies have been reported before, as the author mentioned in line 60 to line 71, however, the extensive mutant protein structures, and discussions they have inspired meet the criterion for novel.)

Completeness: Excellent—this is a Herculean effort.

Reliability: good

Overall: Publication is recommended—with these minor changes/suggestions incorporated.

Questions and Comments:

1. As the audience of Nature Comm. is expected to be general, the authors should consider a general explanation for the design of the SDM. Why did the author choose the specific amino acid (A, D, S, Q) as the mutant acid? Is the SDM designed? If so, what is the expectation?
2. Again, for the general audience, the figures are needed but it is difficult to maneuver between them. Figure 4 should likely be Figure 1 as it expresses the overall strategy. Figure 6 asks the reader to distinguish between a light blue and pale violet. This is just too much work for the reviewer/reader. Please rethink this figure—make it better or relegate to SI. I am sure the authors are making a point, but this reviewer gave up on it. Chem Draw cartoons would work better in some cases. Maybe in Figure 4. Show the essence of the mutants in graphic design, rather than just lifting them from the x-ray structures. Too much similarity and difficult to see the point. Also the game-plan might be better expressed if the authors used some chemical structures rather than all x-ray downloads.
3. There is not a dramatic decrease in H₂ production activity for E282D, E144D, R286A, R148A. Do the authors have other experimental evidence for the H⁺ transfer role of E282, E144, R286, R148?
4. Line 99, "In general, conservative amino acid exchanges had less 100 dramatic effects than non-conservative substitutions."

This sentence need reference; also I think the authors meant "conserved" rather than "conservative".

1. Line 208, "Crystallization of HydA1 exclusively succeeded for apo-protein which carries the [4Fe]H-cluster but lacks the [2Fe]H-site (Supplementary Fig. 6). However, comparisons of crystal structures of Cpl apo and holo-protein with apo-HydA1 demonstrated that a lack of cofactor does not affect the configuration of the putative PT-pathway"

The lack of cofactor does not affect the configuration of PT-pathway, however, it will affect the H₂ evolution activity. How do the authors guarantee that during the H₂ evolution activity reaction, the enzyme contains the cofactor? If the enzyme does not have the cofactor, it is reasonable that the mutant has lower catalytic ability. In other words, Is the catalytic ability decrease due to partial loss of catalytic cofactor?

2. Line 220, "According to their inter-residue distance, R286Cpl might function as a salt bridge partner of deprotonated E282Cpl"

What is the specific number of inter-residue distance?

3. Line 245 "(iii) In variant S319ACpl, the carboxyl group of E282 is slightly shifted outward

probably due to the removal of the hydroxyl group at position 319 that served as H-bond donor (Fig. 4)."

This needs more description.

4. Line247 "Although its distance to the carboxyl group of E279 amounts to 5.8 Å, the presence of Wat276 may explain the dramatically diminished yet detectable H₂ release activity of variant S319ACpl"

Can the 5.8 Å away water transfer H⁺ from E282 to S319A?

"

Reviewer #2 (Remarks to the Author):

The manuscript reports a very detailed analysis of the proton transfer pathway in [FeFe] hydrogenases.

Although the subject has already been tackled in several works, the merit of this paper is the huge amount of experimental evidences obtained via crystallographic structures and FTIR analysis on a series of variants of the pathway, on the two mainly studied enzymes of this type, namely Cpl and HydA1. The impressive experimental work will also supply data for the analysis by the scientific community, especially on the crystal structures provided.

Few minor points might still need to be clarified:

1) the role of Serine 319 (or 189) in proton pathway is still a bit elusive, given the high turnover of the serine to alanine mutant, and also an FTIR spectrum that closely resembles the WT. The fact that a water molecule is observed in this mutant as a bridging between the two Glutamic acid partially explains the data, but also suggest that water might have a role also in the WT in that position: is there any evidence of a partial occupancy of water molecules also in the WT? How do the authors suggest that the transfer is mediated by this residue? The point is slightly less detailed in the paper than the other part of the pathway

2) in figure 5 the pK_a of this same Serine is not reported: is there any reason why? As a suggestion, if possible, also any modulation of water pK_a in the positions where water is bridging the proton transfer would be very useful to report. As a minor remark on the same figure 5, the distance is not very clearly reported between E279 and S219

3) indicating the occupancy of the water molecules involved in the pathway in all variants (maybe in the SI) would be highly useful in quickly evaluating possible dynamic exchange of water in the pathway itself, also it will enable the analysis in the future of other variants and other wild type enzymes. The same holds for a brief discussion on the possible H-bonding network that might stabilise the water and favour its positioning in the variants

4) the authors have missed to discuss the relation to their work of another recent paper on the subject by the group of Hegg. (Biochemistry. 2016 Jun 7;55(22):3165-73).

5) as a minor point in the discussion, on line 177 the authors report as surprising the fact that the C299D (or C169D) variants cannot be trapped and accumulated in the H_{hyd} state. This, given the high similarity, from a functional point of view, of this variant to the WT (still 50% activity retained in both evolution and uptake), as well as the fact that the acidic pH, which in the WT favours the H_{hyd} state, in the C299D variant is already positively affecting the turnover, does not seem very surprising to me.

Reviewer #3 (Remarks to the Author):

The authors present a substantial body investigating the putative proton transfer pathway to and from the H-cluster of [FeFe]-hydrogenases Cpl and HydA1 using a combination of site directed

mutagenesis, activity assays, IR-spectroscopy and x-ray crystallography. The experiments are well conceived and performed to a high standard and the authors congratulated for their efforts. I believe the work is suitable for publication in Nature communications once the following amendments are made.

Could the authors speculate on why the equivalent mutations E282Q and E144Q in Cpl and HydA1 result in an order of magnitude difference in activity?

The high pKa of E279 is said to reflect its hydrophobic environment yet there are two water molecules immediately adjacent to it (shown in all the figures) and it is also in hydrogen bonding distance of S319, could the authors clarify how the pKa is modulated.

The electron density and temperature factors of E282/E279 are used, quite reasonably, as evidence of a single conformation of these residues as opposed to the bi-stable conformations described by Long et al. Have the authors considered that the residues may be trapped in a single conformation since all the proteins are thought to represent the Hox catalytic intermediate, and short lived alternate conformations could exist during turnover? Possible crystallising reduced the protein with the cluster in the Hhyd could reveal motion of these residues. (I am not suggesting this is a requirement for publication, but some discussion would strengthen the paper further).

Also proton transfer is described as a step-by-step process with a proton hopping from residue to residue. Could a more concerted mechanism occur with one proton leaving a residue as another bonds to replace it?

The blocking of proton transfer invoke the presence of a Zundel ion, is the hydrogen bonding network to the protein compatible with the geometry the water molecule would be required to adopt in order to form such an ion? Also is there any evidence in the IR for the presence of this type of ion?

Minor edits.

line 92 – (Fig 1b) should this be (Fig 1a)?

line 102-103 The text states variants R286A and E28A only mildly affected activity with values ~30%, yet the graphs in Figure 1 suggest ~90% activity for R286A and ~60% for E282A, which is correct?

Lines 147-149 – Text refers to Cpl/HydA1 and their SDM mutants in supplementary figure 2, but only data for HydA1 mutants appears to be presented. The text should be amended to clarify this.

Line 156 – should read “in the presence”

Figure 3 – Lines to indicate the wavelength of each peak in the spectra (similar to supplementary figure 4) would clarify the information presented.

Line 225 – should read “significantly fewer water molecules”

line 249 – should read “diminished yet still detectable”

Lines 296-312 – Hydrogen bond lengths reported in the text do not match the values shown in figure 5, this should be corrected.

Figure 5- For clarity, the distances in blue should be more explicitly linked to the atomic spacings they represent.

Line 349 – the Zundel ion formula should be H₅O₂

Supplementary table 4 – CC1/2 values for each data set should be reported.

Also the Cpl-C229D contains 1305 water molecules, significantly more than any another structure, is this correct?

Reviewer #4 (Remarks to the Author):

As described in the manuscript, the fast catalytic turnover rates of [FeFe]-hydrogenases must have an exceptionally efficient proton-transfer (PT) pathway to shuttle protons bulk water and catalytic center. Although there exist many proposals for the proton transfer mechanism in theory, this manuscript is the first ever to experimentally prove one of the the proposal as the real mechanism in Cpl and HydA1 [FeFe]-hydrogenases and their site-directed mutagenesis. The combination of dynamics, FTIR and crystallography clearly demonstrated the proposed PT-pathway with solid experimental proof. In particular, the FTIR spectra do support the proton transfer proposal well. I therefore recommend the manuscript to be published in Nature Commu. but With some following miner questions and revisions:

1) At the beginning, it could be better to explain what reserved and non-reservd residues are for non-biochemistry readers, but I leave this issue to the authors or editor to make the call;

2a) line 92, it is better to add "(Cpl)" after number "860" to make it clear;

2b) line 93, it is better to add "HydA1" after number "2600";

3) line 96, I believe the last word should be "(A or S)" - NOT only (A);

4) line 108, it is better to have some way to label the lower and upper boundaries of the white gap; or add the numbers for the pairs of lowest bars which surpass the gap;

5) line 148, when discussing Fig SI2, it is better to add the text comment "in relative scale" (as shown in the caption of the Fig SI2);

6) line 151-154, it is better to add more writing from SI to here to better discuss the issue of H₂ oxidation;

7) line 174, again it is better to label the number for S189A in Figure 1b and mentioned it in text here, so the discussion becomes clearer;

8) line 179, FIG 3, R148A seems (the only one) to have a mixed states plus some H(hyd) state. Please point out and explain why R148A is NOT the same as others;

9) I also suggest to move FIG SI 5 (FTIR) (or part of it) to combine with FIG3 in the main text;

10) line 269-299, this explains the specialty of C299D / C169D and should be moved or repeated in the discussion/mechanism part (not only here);

11) line 285, "to uncover the individual contribution of residues to proton transfer in the putative PT-pathway of [FeFe]-hydrogenases", it seems better (in here or anywhere) to summarize a list to show how each residue affects the proton transfer in each particular way;

12) line 327, authors may consider adding some SI discussions here;

13) line 330, at the end, I think there should be a "," before "it" - I also found similar typos in

other places, please correct them;

14) line 357, "variant C299ACpI, the first deprotonation step ($\text{HhydH}^+ \rightleftharpoons \text{Hhyd} + \text{H}^+$) seems impossible but...", why is it impossible?

15) Authors cited several publications from modern nuclear resonant vibrational spectroscopy, such as the ones in line 565-569. The following are a few recent papers which are directly related to FeFe hydrogenases and thus are worth citing:

a) Pelmeshnikov et al, J. Am. Chem. Soc., 2017, 139 (46), pp 16894–16902

b) Cramer et al, <https://doi.org/10.1002/anie.201805144>

c) Pelmeshnikov et al, <https://doi.org/10.1002/anie.201804601>

Reviewer #1 (Remarks to the Author):

Review for Crystallographic and spectroscopic assignment of the proton transfer pathway in [FeFe]-hydrogenases

Content:

The powerful tool of site directed mutagenesis was used to investigate the PT-pathway of Cpl and HydA1. This resulted in 11 mutants for two [FeFe]-H₂ases, characterized by FTIR and 11 crystal structure analyses as a basis for interpretation of enzyme kinetics in H₂ production and oxidation/proton release activity. The overall review of the design of the diiron hydrogenase enzyme, proton delivery or removal route, is put on very firm ground. In addition it verifies the convergent evolution not only of the active sites, but also of the outer machinery that makes the enzymes so efficient for hydrogen processing.

Overall Evaluation:

Novelty : medium to high (the idea of site-directed mutagenesis of amino acids in PT-pathway is not new; several studies have been reported before, as the author mentioned in line 60 to line 71, however, the extensive mutant protein structures, and discussions they have inspired meet the criterion for novel.)

Completeness: Excellent—this is a Herculean effort.

Reliability: good

Overall: Publication is recommended—with these minor changes/suggestions incorporated.

Questions and Comments:

Question 1

As the audience of Nature Comm. is expected to be general, the authors should consider a general explanation for the design of the SDM. Why did the author choose the specific amino acid (A, D, S, Q) as the mutant acid? Is the SDM designed? If so, what is the expectation?

Answer:

We agree with reviewer 1 that we should outline our SDM strategy here a bit better for clarity reasons and accordingly added the following paragraph:
,For both, Cpl and HydA1, eleven SDM variants were generated to target residues along the putative PT-pathway, applying conservative and non-conservative exchanges (Fig. 1). Conservative exchanges (e.g. E→D) conserve the functional group of the targeted position, but due to other structural differences in the substitute residue, will affect the precise spatial placement and configuration of the functional group. In a highly ordered system such as the well-distanced H-bond chain of an evolutionarily optimized proton transfer pathway this should at least affect the efficiency of the functional aspect. Non-conservative exchanges (e.g. E→Q/A) delete the functional group entirely and therefore prohibit these substitute residues from rescuing the targeted function.'

Question 2:

Again, for the general audience, the figures are needed but it is difficult to maneuver between them. Figure 4 should likely be Figure 1 as it expresses the overall strategy.

Answer:

We thank reviewer 1 for the suggestion to improve the clarity of our manuscript. However, switching Figures 1 and 4 would give away the crystal structure data right from the start. However, to our opinion they should follow the kinetic and IR-spectroscopic data as they provide plausible explanations for some of the unexpected results. To nevertheless enhance the clarity of the entry part, we added the exchanges made for each proton transfer position to subfigure A of Figure 1 and included the following sentence in the figure legend:

,The substitutions applied in this study for individual positions are shown in parentheses below the respective position labels.'

Question 3:

Figure 6 asks the reader to distinguish between a light blue and pale violet. This is just too much work for the reviewer/reader. Please rethink this figure—make it better or relegate to SI. I am sure the authors are making a point, but this reviewer gave up on it. **Chem Draw cartoons** would work better in some cases.

Answer:

We followed the advice of reviewer 1 to prepare Chem Draw cartoons for **Figure 6** instead of showing structure models to focus on the core features which emphasize the main conclusions drawn for each variant. The original figure has been translocated to the SI part as **Supplementary Figure 14**.

Question 4:

Maybe in Figure 4. Show the essence of the mutants in graphic design, rather than just lifting them from the x-ray structures. Too much similarity and difficult to see the point. Also the game-plan might be better expressed if the authors used some chemical structures rather than all x-ray downloads.

Answer:

In case of figure 4 the focus lies on depicting the structural differences within the proton transfer pathways of the examined variants compared to wild-type protein, which we can draw directly from the X-ray structure data, including the impact of made substitutions on inter-residual distances or orientation of individual amino acid residues/H₂O-molecules. As these features can be clearly (and for the sake of credibility should be) derived from the crystal structure data, we find it vital to show the structures and electron density maps here instead of cartoons.

Question 5:

There is not a dramatic decrease in H₂ production activity for E282D, E144D, R286A, R148A. Do the authors have other experimental evidence for the H⁺ transfer role of E282, E144, R286, R148?

Answer:

We agree with reviewer 1 that the decrease in H₂ production activity for the indicated variants of the two surface exposed residues (E282A/D and R286A and corresponding variants of HydA1) is comparatively low but there are numerous strong indicators that at least position E144/E282 is vital for proton transfer. Variants E144/E282=>Q and A as well as E282D accumulate H_{hyd} at pH8 like most of the other PT-variants. E144D at least accumulates H_{hyd} to a relatively large fraction compared to wild-type. In addition to that, for H₂ evolution we documented significant shifts in the pH-activity profiles of E144/E282=>Q/A and D of 0.5-1.5 pH units towards the lower pH range. In case of variants R148/R286=>A we also see slight changes in the pH activity profile and a more pronounced fraction of H_{hyd} under H₂ at pH8, similar to E144D. For E144A and R148A we measured a strong decrease of H₂-oxidation activity and deviating pH-activity profiles. However, given the comparatively low level of impact modulations of R148/R286 have on each of these indicators, we concluded in the main text that the Arginine residue probably modulates the *pKa* and protonation state of E282 and might stabilize the H-bond network of closely situated surface-attached water molecules, rather than directly participating in proton-transfer.

Question 6:

Line 99, "In general, conservative amino acid exchanges had less 100 dramatic effects than non-conservative substitutions." This sentence need reference; also I think the authors meant "conserved" rather than "conservative".

Answer:

We thank the reviewer for the chance to clarify this term. Conservative and non-conservative is correct, see e.g.: French, S. & Robson, B. J Mol Evol (1983) 19: 171. <https://doi.org/10.1007/BF02300754>. We added this reference to our manuscript. Conservative amino acid substitutions are confined to members of the same amino acid subgroup (hydrophobic, polar, acidic, basic...) which carry the same or a similar functional group (such as a carboxylic group in D or E) in their residue. If this functional group is involved in a specific enzymatic feature (such as proton transfer via an H-bond chain) conservative substitutions likely rescue at least a certain fraction of this function while the substituents of a non-conservative exchange don't provide this functional group and are therefore incapable of rescuing the respective enzymatic feature by themselves. As shown in this study, this doesn't necessarily mean the targeted function cannot be rescued by other molecules (such as H₂O-molecules) which

might occupy the space left over by the substitution. See also answer to comment 1.

Question 7:

Line 208, “Crystallization of HydA1 exclusively succeeded for apo-protein which carries the [4Fe]H-cluster but lacks the [2Fe]H-site (Supplementary Fig. 6). However, comparisons of crystal structures of Cpl apo and holo-protein with apo-HydA1 demonstrated that a lack of cofactor does not affect the configuration of the putative PT-pathway” The lack of cofactor does not affect the configuration of PT-pathway, however, it will affect the H₂ evolution activity. How do the authors guarantee that during the H₂ evolution activity reaction, the enzyme contains the cofactor? If the enzyme does not have the cofactor, it is reasonable that the mutant has lower catalytic ability. In other words, Is the catalytic ability decrease due to partial loss of catalytic cofactor?

Answer:

To exclude the possibility that a large fraction of the variants lose their cofactor (activity) under turnover condition (e.g. H₂ evolution), we performed a control experiment (see below).

Specific activity of HydA1-WT, -E144D, -S189A and -E141D were measured using the method described in the manuscript with slight modifications. First, H₂ production activity was measured after incubating the samples for 10 min instead of the 20 min used in the manuscript, (shown as blue columns). Afterwards, the same samples were purged for 7 minutes with Ar to remove the already produced fraction of H₂ before incubating for another 10 min (The complete removal of H₂ was confirmed.). The activity measured with the same sample after the 2nd turn is shown in red columns.

The slight decrease from 1st to 2nd turn by 10-25% measured for both, wild type and variant samples might be referred to a slight O₂ contamination or a decrease in substrate concentration (sacrificial electron donor (NaDT)) during the second incubation period. As the relative decrease is very similar among wt enzyme and variants, even for those, with a very low activity level (S189A and E141D), we can excluding the possibility that the applied substitutions

render the enzyme variants more prone to cofactor loss or degradation compared to WT.

In the following, we further describe that also under non-turnover conditions cofactor is largely retained in most of the variants. Steric side-effects caused by the applied amino acid exchanges, which may result in a significant loss of enzyme activity due to incomplete cluster incorporation, can be excluded for most of the variants by determining the relative cluster content in reference to the total amount of enzyme via FTIR-spectroscopy. After N₂ gassing the amplitude of the lead CO-vibrational signal at 1940 cm⁻¹ in H_{ox} was compared to the amplitude of amide band II (C=O stretching and N-H bending vibrations of the main chain) revealing for most variants a cluster content of >75% compared to wild-type enzyme. Only C169D (and S, however not further discussed here) exhibited a significantly lower H-cluster content of <50% (see Figure below).

In Cpl structures, all variants showed an occupancy for the [4Fe4S]_H cluster between 95 and 100% except for C299A, which for the cubane sub-cluster exhibited an occupancy level of 90%. As for the occupancy of the [2Fe]_H sub-cluster, most of the Cpl variants are fully equipped. E282Q and E279Q variants showed a slightly lower [2Fe]_H occupancy of 85% and 96%. Variants of C299 showed the lowest occupancy levels for [2Fe]_H, with 70% and 65% in case of C299A and C299D, respectively. So at least for the C169D/C299D variants the lower [2Fe]_H content could be responsible for some of the loss of enzymatic activity. However, for this variant, we still measured with both, Cpl and HydA1, a comparatively high level of residual catalytic activity, which suggests a certain instability and loss of H-cluster integrity over time, probably due to the steric influence of the larger substitute residue. An earlier data set from older crystals of variant C299D_{Cpl} (not shown in the manuscript) even indicates that the carboxy-group sooner or later might even react with the azadithiolate bridge to form a carboxamide-group, which would render the enzyme inactive and might support cofactor degradation.

To clarify that our data unimonously show that the variants examined here are largely occupied with the entire H-cluster, we added the following sentence to the manuscript: *,and the the H-cluster was largely present for all Cpl variants (Supplementary Table 5).*

Influence of mutagenesis on H-cluster occupancy in HydA1 verified via FTIR. a: All the spectra were recorded by purging the enzymes with N₂ at pH8 to accumulate H_{ox}, and they were normalized in signal size in reference to the amide II band at 1540 cm⁻¹. The interesting region being characteristic for vibrational signals of the diatomic H-cluster ligands was marked by a green rectangle. **b:** Here, the CO-ligand signals of all HydA1 mutagenesis variants (color code corresponds to the one used for Fig. 1) are individually compared to wild type (black line).

Question 8:

Line 220, "According to their inter-residue distance, R286C_{pl} might function as a salt bridge partner of deprotonated E282C_{pl}" What is the specific number of inter-residue distance?

Answer:

The distance between the carboxy group of E282_{C_{pl}} and the the guanidinium group of R286_{C_{pl}} is 2.8Å (see Fig. 5) which would qualify them to be potential salt bridge partners. To clarify this, we slightly revised the indicated sentence:

,According to their close inter-residue distance of 2.8 \AA , $R286_{Cpl}$ might function as a salt bridge partner of deprotonated $E282_{Cpl}$ (Fig. 4, right and Supplementary Figs. 6-7) and thus could be involved in the PT mechanism.

Question 9:

Line 245 “(iii) In variant S319ACpl, the carboxyl group of E282 is slightly shifted outward probably due to the removal of the hydroxyl group at position 319 that served as H-bond donor (Fig. 4).” This needs more description.

Answer:

To clarify this point we revised this passage as given below:

'In variant $S319A_{Cpl}$, the carboxyl group of $E282_{Cpl}$ is slightly shifted outward probably due to the lack of the hydroxyl group at position 319_{Cpl} . In wild type enzyme $S319_{Cpl}$ acts as H-bond donor to $E282_{Cpl}$ and drags its carboxyl group further into the proton transfer pathway (Fig. 4). In $S319A_{Cpl}$ an unprecedented water molecule (Wat285 in chain B) is located between E282 and A319. At the same time the nearby water molecule Wat873 of wild type Cpl (chain B) is missing here, suggesting a translocation, enabled by the unoccupied space of the missing hydroxyl-group of position 319 and the slight outward shift of E282 (see Supplementary Fig. 12c). Although its distance to the carboxyl group of E279 amounts to 5.8 \AA , the presence of Wat285 may explain the dramatically diminished, yet still detectably H_2 release activity of variant $S319A_{Cpl}$ (6%).

Concerning the additional H_2O molecule in the proton transfer pathway of $S319A_{Cpl}$, please also see our answer to question 3 of reviewer 2.

Question 10:

Line247 “Although its distance to the carboxyl group of E279 amounts to 5.8 \AA , the presence of Wat276 may explain the dramatically diminished yet detectable H_2 release activity of variant S319ACpl”

Can the 5.8 \AA away water transfer H^+ from E282 to S319A?

Answer:

We admit that the 5.8 \AA of long distance between the water and E279 excludes a direct proton transfer. However, we assume that some highly mobile water molecules (inaccessible by X-ray crystallograph) may bridge the big gap (indeed we observe very low additional density, not supporting placement of ordered water molecules). Certainly, the rate was dramatically slowed down, which agrees well with ~5-15% of WT activity we measured.

Reviewer #2 (Remarks to the Author):

The manuscript reports a very detailed analysis of the proton transfer pathway in [FeFe] hydrogenases. Although the subject has already been tackled in several works, the merit of this paper is the huge amount of experimental evidences obtained via crystallographic structures and FTIR analysis on a series of variants of the pathway, on the two mainly studied enzymes of this type, namely Cpl and HydA1. The impressive experimental work will also supply data for the analysis by the scientific community, especially on the crystal structures provided.

Few minor points might still need to be clarified:

Question 1:

The role of Serine 319 (or 189) in proton pathway is still a bit elusive, given the high turnover of the serine to alanine mutant, and also an FTIR spectrum that closely resembles the WT. The fact that a water molecule is observed in this mutant as a bridging between the two Glutamic acid partially explains the data, but also suggest that water might have a role also in the WT in that position: is there any evidence of a partial occupancy of water molecules also in the WT? How do the authors suggest that the transfer is mediated by this residue? The point is slightly less detailed in the paper than the other part of the pathway

Answer:

This is a valid point. However, we checked the electron density map of the WT structure (4XDC) which has a relatively high resolution (1.63 Å), but there are no hints for the partial occupancy of a water molecule in this location. Concerning the role of serine in the proton transfer mechanism, we agree with reviewer 2 that it is less detailed addressed in our manuscript. In mutagenesis studies of other systems (bacterial reaction center: Paddock et al. 1990 PNAS; Monooxygenase Hydroxylase: 10.1021/ja1063795; Cytochrome c Oxidase: 10.1021/bi100749p), the unusually high *pKa* of serine/threonine was shown to be vital for proton-transfer. The high *pKa* renders it constantly 'protonated' close to physiological conditions, which is distinctive from other residues of the pathway, such as Glu, which can adopt two different protonation states. Its direct contribution to proton transfer has only recently been outlined by Salna et al (doi: 10.1038/nchem.2527). They suggested that high-*pKa* residues like serine support proton transfer by a deep tunnelling process. In our study, we showed that the S→A/D mutants are largely inactive, and exhibit different pH titration behaviors. Most importantly, the HydA1-S189D variant under H₂ gassing at pH8 dominantly adopted the H_{hyd} state. Altogether, this allows us to conclude that S189_{HydA1}/S319_{Cpl} is indeed involved in proton-transfer, presumably by mediating protons between E279_{Cpl} and E282_{Cpl} via deep-proton-tunneling.

Question 2:

in figure 5 the pKa of this same Serine is not reported: is there any reason why? As a suggestion, if possible, also any modulation of water pKa in the positions where water is bridging the proton transfer would be very useful to report. As a minor remark on the same figure 5, the distance is not very clearly reported between E279 and S219

Answer:

The *pKa* of serine is very high (15-17) compared to other functional residues, such as Glu and Arg. Unfortunately, as it is rarely considered to be deprotonated under physiological conditions, serine residues are not covered by the 'PROPKA' analysis, neither are the *pKa* values of bound water molecules. The distance between E279 and S319 is 3.6Å. We have slightly revised Fig. 5 to make the distances more discernible.

Question 3:

Indicating the occupancy of the water molecules involved in the pathway in all variants (maybe in the SI) would be highly useful in quickly evaluating possible dynamic exchange of water in the pathway itself, also it will enable the analysis in the future of other variants and other wild type enzymes. The same holds for a brief discussion on the possible H-bonding network that might stabilise the water and favour its positioning in the variants.

Answer:

Following the advice of reviewer 2, we prepared additional figures showing the H-bond networks that stabilize water molecules directly and/or indirectly (e.g. by being in the coordination sphere of residues of the proton transfer pathway) involved in the proton transfer of wild type Cpl as well as for additional or translocated water molecules, present within the pathway of individual variants examined here (S319A; E282A; C299A). Furthermore, the corresponding b-factors for the H₂O molecules included in this structural analysis have been added to **Supplementary Table 8**. The structural data gained from this detailed analysis nicely supports and elaborates our interpretations concerning the individual features of the respective variants. The following figures and legends were added as **Supplementary Fig. 12** and **Fig. 13** to the Supplementary Information:

Supplementary Figure 12 | H-bond networks stabilizing protein bound water molecules which participate in proton transfer in Cpl wild type and S319A. Only H-bond markers for distances $\leq 4\text{\AA}$ are presented. H-bond distances between water molecules or to residues of the proton transfer pathway are depicted in black. H-bond contacts of H₂O-molecules to the protein environment are presented in yellow. a: H-bond networks for W862, W1120

(carbon atoms of amino acids with participating main chain and side chain groups are marked in green) and W873 (carbon atoms of amino acids with participating main chain and side chain groups are marked in grey) in Cpl wild type. b: H-bond network stabilizing W735 in S319Acpl shifted from its original position in wild type protein (W873) as a consequence of the amino acid exchange. Stick structures of wild type Cpl (4XDC; black) and S319Acpl (cyan) have been superposed to compare the original and translocated position of the water molecule with the H-bond network changing from five potential contacts to main and side chain groups of E282, G560, E278 and H569 to six contacts with E282, E279, A319, A321 and S320. c: Enlargement of the two water positions in Cpl wild type and S319Acpl demonstrating the steric hindrance of W873 translocation in wild type (black) due to the presence of the hydroxyl group of S319 and the conformation of E282. Its carboxy group is slightly twisted inwards as a consequence of the H-bond contact with S319.

Supplementary Figure 13 | H-bond networks stabilizing additional protein-bound water molecules in the proton transfer pathways of Cpl variants E282A and C299A. Only H-bond markers for distances $\leq 4\text{\AA}$ are presented. Stick structures of wild type Cpl (4XDC; black stick structures and water molecules presented as spheres) and E282Acpl (A) (mint colored green stick structures and water molecules, presented as spheres) or C299Acpl (B) (light orange and water molecules) have been superposed to compare the positions of the

translocated water molecule in both proteins and additional H₂O molecules are depicted in red (a) and green (b), respectively. Blue spheres depict water molecules which are located at the protein surface. H-bond distances between water molecules or to residues of the proton transfer pathway are depicted in black. H-bond contacts of H₂O-molecules to the protein environment are presented in yellow. Residues and water molecules are labeled in corresponding colors, with the substitute amino acid of each variant marked by a yellow background. W717 and W974 of E282ACpl are coordinated by H-bonds to main chain and side chain groups of S319, S320, A321 and surface water molecules W1030 and W942. The additional H₂O molecule in C299ACpl (W962) exhibits five potential H-bond contacts with T297, A299, W1021, W1120 and the azadithiolate ligand of the [2Fe]H cluster.

Question 4:

The authors have missed to discuss the relation to their work of another recent paper on the subject by the group of Hegg. (Biochemistry. 2016 Jun 7;55(22):3165-73).

Answer:

The indicated study of Hegg and coworkers (Cornish et al., 2016) suggested that the exchange R286L would actually lead to an increase of enzyme activity compared to wild-type Cpl. In a collaborative effort our groups exchanged our respective protein variants (R286L of the Hegg group → our group and R286A and R286L from our group → Hegg's group). With the samples from the other group, both could verify the results of the other group compared to the corresponding wild type protein sample. However, we also realized that the wt protein samples received from the Hegg lab only showed a minimal fraction of the activities we measured for our wt protein (2.3%: $1.0 \mu\text{mol mg}^{-1} \text{s}^{-1}$ v.s. $2576 \mu\text{mol mg}^{-1} \text{min}^{-1}$) even though the measurements were done under different temperatures, 25 and 37°C respectively for the Hegg lab and our lab (we also measured the activities at 25°C. They were roughly half of the value measured at 37°C). Furthermore, the activity ($1.0 \mu\text{mol mg}^{-1} \text{s}^{-1}$) reported by the Hegg group in 2016 (Cornish et al. 2016) is even very low compared with value reported earlier under the very similar condition (1000s^{-1} , equals to $7.9 \mu\text{mol mg}^{-1} \text{s}^{-1}$, Cornish et al. 2011). It turned out that the Hegg group used a different expression and maturation procedure compared to our group which left proteins from the Hegg group insufficiently matured. When subjecting the insufficiently matured samples of the Hegg group to our *in vitro* maturation procedure (Esselborn et al. 2013), we were able to mature their proteins to the same level as our protein samples (nearly 100%). As a consequence however, the activity of R286L was then slightly lower compared to the equally treated wild-type sample of the Hegg group which finally confirmed the results we describe here for R286A. As both groups used different expression and maturation systems the results cannot be compared.

Question 5:

as a minor point in the discussion, on line 177 the authors report as surprising the fact that the C299D (or C169D) variants cannot be trapped and accumulated in the H_{hyd} state. This, given the high similarity, from a functional point of view, of this variant to the WT (still 50% activity retained in both evolution and uptake), as well as the fact that the acidic pH, which in the WT favours the H_{hyd} state, in the C299D variant is already positively affecting the turnover, does not seem very surprising to me.

Answer:

We agree with reviewer 2 that it is not surprising that for C299D/C169D H_{hyd} can't be trapped at pH8 as it is the case for the low active or inactive variants of the proton transfer pathway. However, we expected wild-type like behavior at pH4 (H_{hyd} accumulation) and were surprised that even under high proton pressure we couldn't identify any fraction of H_{hyd}. Reviewer 2 might be right that this could be at least partially due to the fact that for C299D the activity maximum for H₂ evolution is shifted by 1.5 units to the lower pH range (from pH8 to pH6.5) and even at pH5 we can see a nearly 20-fold higher relative activity, compared to wild type. Therefore, we agree with reviewer 2 that the higher residual activity of C169D/C299D in the lower pH range compared to wild type might indeed prevent H_{hyd} accumulation. We thank reviewer 2 for this insightful comment.

We substituted in the introductory word *'Surprisingly'* in the indicated sentence for *'Interestingly'* and added the following section in reference to C169D/C299D to the discussion part:

,This comparatively high level of absolute activity and the fact that both corresponding variants, C169D_{HydA1} and C299D_{Cpl} exhibit significant shifts in their pH-dependent activity optimum to the acidic range, might explain why this type of variant refuses to accumulate H_{hyd} even at pH4, while instead effectively continuing the turnover process, as evident according to the comparatively dominant IR-bands for the reduced states H_{red} and H_{sred} (1890 cm⁻¹ and 1882 cm⁻¹ for C169D) recorded at pH4 (see Supplementary Fig. 4).'

Reviewer #3 (Remarks to the Author):

The authors present a substantial body investigating the putative proton transfer pathway to and from the H-cluster of [FeFe]-hydrogenases Cpl and HydA1 using a combination of site directed mutagenesis, activity assays, IR-spectroscopy and x-ray crystallography. The experiments are well conceived and performed to a high standard and the authors congratulated for their efforts. I believe the work is suitable for publication in Nature communications once the following amendments are made.

Question 1:

Could the authors speculate on why the equivalent mutations E282Q and E144Q in Cpl and HydA1 result in an order of magnitude difference in activity?

Answer:

For the case of the E/Q exchange of the surface exposed glutamate we agree with reviewer 3 that between both enzymes there is an unexpected level of deviation in residual activity. This might fit to differences we observed in respect to the individual crystal structures of both enzymes. In E282Q_{Cpl}, we see two different conformations for Q282, equally distributed, close to the entrance of the pathway (**Supplementary Figs. 6 and 7**). In case of E144Q_{HydA1} we observed an unambiguous conformation for the Gln residue (the raw data for crystal structure of E144Q do exist in our group but they are less well refined and therefore have been omitted in the manuscript). We assume a higher level of flexibility for the residue of Q282 in E282Q_{Cpl} which in some cases might even allow water molecules to come close enough to S319 to enable a slightly higher level of activity compared to the corresponding HydA1 variant. When comparing activity values of corresponding variants in Figure 1B, there is a significant trend for Cpl variants of the more surface exposed PTP-positions (R286 E282 and even S319) to have higher residual activities compared to the corresponding HydA1 variants, which might suggest for Cpl an overall more open or flexible access to the proton transfer pathway which can tolerate variations slightly better than HydA1.

To address this originally neglected aspect in our manuscript we added the following section:

,Further, there is a general trend for substitutions of the more surface exposed PTP-positions (R286 E282 and even S319) in Cpl to have slightly lower impact on enzymatic activity, compared to the corresponding HydA1 variants. It might suggest for Cpl a more open or flexible access to the proton transfer pathway, which can tolerate variations slightly better than HydA1. This is especially obvious by the 20-fold difference in the relative activity of E282Q_{Cpl}, compared to its HydA1 counterpart E144Q.'

Question 2:

The high pKa of E279 is said to reflect its hydrophobic environment yet there are two water molecules immediately adjacent to it (shown in all the figures) and it is also in hydrogen bonding distance of S319, could the authors clarify how the pKa is modulated.

Answer:

While hydrogen bonding usually lowers the pK_a of Glu, the exact pK_a also depends on the type, strength and angles etc. of the hydrogen bonding (doi:10.1002/prot.20660). The main reason for E279_{cpi} reaching a rather high pK_a is its buried environment (rather hydrophobic) with only very limited H-bond partners (Ser 319 and wat826) (the 2nd water is >4.5 Å away from it). The H-bond to Ser319 is even relatively weak, thus implicating the H-bond to Wat826 to be the only significant one. More importantly, the pK_a calculated here with PROPKA (8.6), agrees very well with 8.7, which was calculated for the same residue by another method (alchemical free-energy perturbation)(Long et al. 2014, doi: 10.1021/jp408621r).

Question 3:

The electron density and temperature factors of E282/E279 are used, quite reasonably, as evidence of a single conformation of these residues as opposed to the bi-stable conformations described by Long et al. Have the authors considered that the residues may be trapped in a single conformation since all the proteins are thought to represent the Hox catalytic intermediate, and short lived alternate conformations could exist during turnover? Possible crystallising reduced the protein with the cluster in the Hhyd could reveal motion of these residues. (I am not suggesting this is a requirement for publication, but some discussion would strengthen the paper further).

Answer:

We agree with refree 3 that we cannot rule out the possibility that certain configurations are favored under the rather artificial crystallization and storage conditions of the protein crystals and accordingly added the following sentence to the discussion part:

,However, we cannot rule out the possibility that such differences between theoretical data and structural information are influenced by the non-physiological crystallization conditions or low temperature crystal storage in liquid nitrogen, which might favor a certain configuration for the residues in the proton transfer pathway.'

Question 4:

Also proton transfer is described as a step-by-step process with a proton hopping from residue to residue. Could a more concerted mechanism occur with one proton leaving a residue as another bonds to replace it?

Answer:

We agree with referee 3 that both models would fit here. To clarify this, we added the following paragraph at the indicated position to the discussion part. *This observation would favor a model in accordance with the Grotthus mechanism, based on simultaneous deprotonation/protonation events, according to a 'bucket line' between unalterably orientated residues and water molecules. However, a bi-stability of the glutamate residues under turnover conditions might still be a valid interpretation, as it could also be regarded as switching between two different hydrogen bonding patterns. In both cases the direction of proton transfer would be determined by the redox state of the [2Fe]_H-cluster, product pressure and substrate availability (see Supplementary Fig. 9 and Supplementary discussion).*

To further clarify how the two possible models of proton transfer (bi-stability model with two H-bond patterns and bucket line model) would affect the catalytic mechanism proposed here, we added the following paragraph to the end of the supplementary discussion:

The validity of this 'bi-stability' model strongly depends on the existence of the yet unidentified states $H_{hyd}H^+$ and H_{sred}^ which unlike the experimentally verified states H_{ox} , H_{hyd} and H_{red} exhibit a protonated adt bridge. Only the acceptance of states with a protonated adt bridge would require the assignment of a second H-bond pattern. If the adt bridge merely fulfills a proton shuttle function without the capacity to intermittently bind a proton, this second H-bond pattern can be omitted and a simple 'bucket line' mechanism can be assumed.*

Question 5:

The blocking of proton transfer invoke the presence of a Zundel ion, is the hydrogen bonding network to the protein compatible with the geometry the water molecule would be required to adopt in order to form such an ion? Also is their any evidence in the IR for the presence of this type of ion?

Answer

The distance between two oxygen atoms in a Zundel ion has been defined to be 2.39-2.43 Å (doi: 10.1021/ja9101826) while in the structure of 4XDC, this distance between Wat826 and Wat1120 amounts to 2.7-2.8 Å (in 3C8Y, it's 3.0 Å). However, likewise larger distances have been tolerated in earlier studies such as in case of Spassov and coauthors (doi: 10.1006/jmbi.2001.4902) who assigned Wat403 and Wat404 of bacteriorhodopsin (PDB 1c3w) to form a Zundel ion although their distance has been measured to be 2.8 Å. However, as we checked our IR data again and couldn't find spectroscopic evidence for the

presence of Zundel ion, we decided to revise the term Zundel ion to *„Zundle-ion-like configuration“*.

Minor edits:

Question 6:

line 92 – (Fig 1b) should this be (Fig 1a)?

Answer:

As we have now indications for the applied mutagenesis in both partial figures, therefore we changed the phrase from (Fig.1b) to (Fig.1).

Question 7:

line 102-103 The text states variants R286A and E28A only mildly affected activity with values ~30%, yet the graphs in Figure 1 suggest ~90% activity for R286A and ~60% for E282A, which is correct?

Answer:

Actually, we intended to refer to all four variants (E282D_{Cpl}, C299D_{Cpl}, R286A_{Cpl} and E282A_{Cpl}) here, which would cover an activity range between 30% and 90% and therefore ≥30% would be correct. But we agree, that this sentence might be rather unfavorably phrased. We therefore changed it to:
„Variants E282D_{Cpl}, C299D_{Cpl}, R286A_{Cpl} and E282A_{Cpl} were only mildly affected and showed residual activities between 30% and 90%.“

Question 8:

Lines 147-149 – Text refers to Cpl/HydA1 and their SDM mutants in supplementary figure 2, but only data for HydA1 mutants appears to be presented. The text should be amended to clarify this.

Answer:

We thank reviewer 3 for pointing out this unclarity. We accordingly revised the text as follows:
„As shown in Fig. 2 and Supplementary Fig. 2‘. Cpl variants are presented in Fig. 2, while corresponding data for HydA1 variants are presented in Supplementary Fig. 2.“

Question 9:

Line 156 – should read “in the presence”

Answer:

The phrase has been changed according to the suggestion of reviewer 3.

Question 10:

Figure 3 – Lines to indicate the wavelength of each peak in the spectra (similar to supplementary figure 4) would clarify the information presented.

Answer:

As suggested by reviewer 3, we added the the wavenumbers to each significant peak in the spectra of Fig 3.

Question 11:

Line 225 – should read “significantly fewer water molecules”

Answer:

The phrase has been changed according to the suggestion of reviewer 3.

Question 12:

line 249 – should read “diminished yet still detectable”

Answer:

The phrase has been changed according to the suggestion of reviewer 3.

Question 13:

Lines 296-312 – Hydrogen bond lengths reported in the text do not match the values shown in figure 5, this should be corrected.

Answer:

We thank reviewer 3 for pointing out this inconsistency:

We corrected the distance values given in the text according to the distances measured in the wild type structure (pdb-code: 4XDC) presented in Fig. 5.

Line 300, 3.4 and 3.3 has been corrected to 3.5 and 3.2 Å

Line 304, 2.6 has been corrected to 2.5

Line 306, 3.7 has been corrected to 3.6

Question 14:

Figure 5- For clarity, the distances in blue should be more explicitly linked to the atomic spacings they represent.

Answer:

We followed the advice of referee 3 and enhanced the line width that indicate the distance labels.

Question 15:

Line 349 – the Zundel ion formula should be H₅O₂⁺

Answer:

We thank referee 3 for pointing out this error. H₅O₃⁺ has now been corrected to H₅O₂⁺

Question 16:

Supplementary table 4 – CC1/2 values for each data set should be reported.

Answer:

We have now added CC1/2 values for each data set to Supplementary table 4.

Question 17:

Also the Cpl-C229D contains 1305 water molecules, significantly more than any other structure, is this correct?

Answer:

We thank reviewer 3 for pointing out this peculiarity. We realized that for the Cpl-C299D data set in one refinement cycle with phenix, an unusually high number of water molecules was assigned near the protein surface during the 'water update refinement'. Most of the water molecules are located at distances larger than 4.5 Å relative to the protein surface. We did further refinement to fix this problem (the refined structure now provides 1009 water molecules) and re-deposited the coordinates and also updated corresponding figures and tables affected by the changes (Fig.4, Supplementary Figs. 6, 7, Supplementary Tables 2, 4 and 5).

However, the relevant structure parts (including proton transfer pathway and H-cluster) weren't affected by the additional refinement.

Reviewer #4 (Remarks to the Author):

As described in the manuscript, the fast catalytic turnover rates of [FeFe]-hydrogenases must have an exceptionally efficient proton-transfer (PT) pathway to shuttle protons bulk water and catalytic center. Although there exist many proposals for the proton transfer mechanism in theory, this manuscript is the first ever to experimentally prove one of the the proposal as the real mechanism in Cpl and HydA1 [FeFe]-hydrogenases and their site-directed mutagenesis.

The combination of dynamics, FTIR and crystallography clearly demonstrated the proposed PT-pathway with solid experimental proof. In particular, the FTIR spectra do support the proton transfer proposal well. I therefore recommend the manuscript to be published in Nature Commu. but With some following minor questions and revisions:

Question 1:

At the beginning, it could be better to explain what reserved and non-reserved residues are for non-biochemistry readers, but I leave this issue to the authors or editor to make the call;

Answer:

We thank reviewer 4 for pointing out this lack of clarity. For the response I would kindly refer to our answers to comment 1 and 6 of reviewer 1.

Question 2:

line 92, it is better to add "(Cpl)" after number "860" to make it clear;

Answer:

We added (Cpl) as suggested.

Question 3:

line 93, it is better to add "HydA1" after number "2600";

Answer:

We added 'HydA1' as suggested.

Question 4:

line 96, I believe the last word should be "(A or S)" - NOT only (A);

Answer:

Indeed, we thank reviewer 4 for pointing this out. We added 'or S' as suggested.

Question 5:

line 108, it is better to have some way to label the lower and upper boundaries of the white gap; or add the numbers for the pairs of lowest bars which surpass the gap;

Answer:

We now separated lower and upper part of the discontinuous scale bar by a black line.

Question 6:

line 148, when discussing Fig S12, it is better to add the text comment "in relative scale" (as shown in the caption of the Fig S12);

Answer:

To clarify that we present relative values in Supplementary Figures 2 and 3 (just as in Fig 2), we added the following sentence to the captions of Supplementary figures 2 and 3:

,Relative values correspond to % of maximum activity obtained throughout the entire pH gradient.'

Question 7:

line 151-154, it is better to add more writing from SI to here to better discuss the issue of H₂ oxidation;

Answer:

Following the advice of reviewer 4 we translocated Supplementary discussion 1 at the indicated position to the main text:

*,It is known from electrochemistry experiments with different [FeFe]-hydrogenases that the enzyme exhibits an increasing H₂-oxidation rate with increasing buffer pH². Accordingly, the pH-range used in this assay was extended to pH 10. As shown in **Supplementary Figure 3**, H₂-uptake activity of native HydA1 generally enhanced with decreasing H₃O⁺-concentrations, exhibiting a nearly five-fold rate-increase per pH unit between pH 6 and 8. After an intermediate drop between pH 8 and 9 the H₂-oxidation activity further increased to nearly 20.000 μmol H₂ mg⁻¹ min⁻¹ between pH 9 to 10. The absolute activities of site directed mutagenesis variants were significantly diminished compared to wild type HydA1, reaching a pH-optimum of at best 6% (R148A). Up to pH 9, variant R148A_{HydA1} showed a wild type-like trend for the pH-activity profile of relative H₂-oxidation activity, while instead of a second increase between pH 9 and 10, the activity strongly declined, suggesting that this second increase is connected with the titration of the guanidine base of R148. The pH-activity profile of E144A_{HydA1} was quite similar to wild type HydA1, despite a flattening out of the local maximum at around pH 8. The latter suggests that this local activity maximum is depending on the presence of the surface exposed Glu residue. S189A and C169D only showed single activity peaks at pH 8 or 9, respectively with very low H₂-uptake rates of 1-2%, compared to wild type.'*

Question 8:

line 174, again it is better to label the number for S189A in Figure 1b and mentioned it in text here, so the discussion becomes clearer;

Answer:

While we are not quite sure if adding an asterisk to the activity bar of S189A would really help to clarify our point, we nevertheless agree that it might be helpful to indicate the data of Fig.1B here. We accordingly added *,(see Fig.1b)'* to the sentence.

Question 9:

line 179, FIG 3, R148A seems (the only one) to have a mixed states plus some H(hyd) state. Please point out and explain why R148A is NOT the same as others;

Answer

The fact that the pH-dependent H₂ production activity is only mildly affected by the R286A exchange and E282 is already a surface exposed residue being in contact with surface bound H₂O-molecules suggests that R286 isn't directly participating in the H-bond chain of proton transfer positions. As R286 exhibits a rather high pKa (12.4) it also seems to be permanently protonated. The activities for H₂-production and especially for H₂-uptake at higher pH values were nevertheless decreased by the R→A exchange, although the rest of proton transfer pathway is structurally intact. We refer this behavior to two putative functional aspects of R286. First, the loss of the arginine-residue in variant R286A destabilizes the H-bond network at the surface preventing H₂O molecules from being coordinated between the His-residues and E282. The crystal structure of position R286 in wild-type Cpl (pdb: 4XDC) suggests that the arginyl residue is involved in an extensive H-bond network which includes the carboxy-group of E282, His residues H565 and H569 and a number of surface bound H₂O molecules which are kept at H-bond distance to this first Glu residue of the PTP. In the same region significantly less water molecules were observed in the crystal structure of variant R286A. As shown in **Supplementary Figure 8** a pathway of four H₂O molecules lining up between both His-residues and leading up to E282 and R286 seems to be lacking in R286A. Still H₂O molecules can be found in R286A which are located in H-bond distance to E282, thus enabling proton exchange between the otherwise intact H-bond chain in the PT-pathway and bulk water which correlates with the high residual activity of the variant. However, the lower complexity of the H-bond network might nevertheless affect e.g. the influence of the two His-residues on the pKa or the efficiency of E282 to function as a proton shuttle between surface H₂O and the PTP. Another aspect refers to the high probability that R286 forms a salt bridge contact to the deprotonated carboxy-group of E282. According to the pH-titrations described in **section 3.3** the protonated form of R286 which is very likely to dominate at pH8 would support H⁺-reduction (see broken yellow arrow in the left panel of Fig. 4.1) while H₂ uptake activity increases when R286 is deprotonated at pH values of 10 and increasing (see Fig. 3.8). Protonated R286 could stabilize the deprotonated form of the carboxy group of E282 which again would allow to effectively shuttle protons to the proton transfer pathway from the nearby network of H₂O molecules thus, supporting H⁺ reduction activity. A titration to higher pH values such as ≥pH 10 would lead to the deprotonation of R286. Under these conditions, R286 would indeed represent a perfect acceptor group for protons leaving the PTP. This might explain the increase in H₂-oxidation activity from pH 9 to 10 which is not determined if the arginyl residue of R286 is missing as in R286A. In agreement

with this, recent cyclic voltammetry experiments with HydA1 under increasing buffer pH (pH 5, 7 and 9) showed that proton reduction current which was demonstrated to be high in a pH-range from 5 to 7 basically vanishes at a pH of 9 [Lampret et al., 2017].

Summing up, our interpretation of the data gained in our study would rather assign R148/R286 as a helper position to modulate/fine-tune the pK_a and protonation state of the first/last proton transfer position E282 by its tendency to pH-dependently force it into a salt-bridge contact and to strengthen the communication between proton transfer pathway and bulk water by stabilizing the surface H₂O-network near the entrance of the pathway.

Question 10:

I also suggest to move FIG SI 5 (FTIR) (or part of it) to combine with FIG3 in the main text;

Answer:

We agree with reviewer 4 that combining Fig 3 and S5 would enhance the comprehensibility and clarity of our data. We accordingly deleted Fig. S4 in the Supplementary Information, while adding it as a sub-figure to Fig. 3.

Question 11:

line 269-299, this explains the specialty of C299D / C169D and should be moved or repeated in the discussion/mechanism part (not only here);

Answer:

We agree with reviewer 4 and accordingly added the following sentence, addressing the specialty of variants C169D/C299D, to the corresponding paragraph of the discussion part:

,This comparatively high level of absolute activity and the fact that both corresponding variants, C169D_{HydA1} and C299D_{Cpl} exhibit significant shifts in their pH-dependent activity optimum to the acidic range might explain why this type of variant refuses to accumulate H_{hyd} even at pH4, while instead effectively continuing the turnover process as evident according to the comparatively dominant IR-bands for the reduced states H_{red} and H_{sred} (1890 cm⁻¹ and 1882 cm⁻¹ for C169D) recorded at pH4 (see Supplementary Fig. 4).'

Question 12:

line 285, "to uncover the individual contribution of residues to proton transfer in the putative PT-pathway of [FeFe]-hydrogenases", it seems better (in here or anywhere) to summarize a list to show how each residue affects the proton transfer in each particular way;

Answer:

According to the suggestion of reviewer 4, we prepared a table summarizing the specific role of each position of the PT-pathway for the proton transfer process as far as can be derived from our study (see **Supplementary Table 9**).

Question 13:

line 327, authors may consider adding some SI discussions here;

Answer:

Although we agree with reviewer 4 that the clarity of our model proposed at the end of this study would profit from integrating the supplementary discussion into the main text, we fear that as a consequence, the main text would exceed the text size limit. We further find it hard to translocate only parts of the model from the Supplementray discussion into the main text as it is strictly based on arguments which are logically and consecutively build up on one another and therefore cannot be presented independently from each other.

Question 14:

line 330, at the end, I think there should be a "," before "it" - I also found similar typos in other places, please correct them;

Answer:

As suggested by reviewer 4, we revised the comma placement throughout the entire manuscript.

Question 15:

line 357, "variant C299ACpl, the first deprotonation step ($\text{HhydH}^+ \rightleftharpoons \text{Hhyd} + \text{H}^+$) seems impossible but...", why is it impossible?

Answer:

At a first glance it appeared impossible, as the first acceptor residue of the proton transfer pathway (thiol-group) near the active site is missing, whose H^+ -transfer function can't be rescued by the substitute residue (Ala). Without knowing the x-ray structure data it would be counter-intuitive to assume that even the first deprotonation step might occur, which would render the enzyme incapable of reaching the H_{hyd} state. The electron density map however suggests an additional water molecule (Wat962) to occupy the space of the missing thiol group. Given the close distance to the adt headgroup, it should be

capable to execute the first deprotonation of the $[2\text{Fe}]_{\text{H}}$ site after H_2 heterolysis and thus to support H_{hyd} accumulation. By this sentence we simply intended to emphasize how x-ray structure data can provide very plausible explanations for the otherwise unexpected behavior of an enzyme variant.

Question 16

Authors cited several publications from modern nuclear resonant vibrational spectroscopy, such as the ones in line 565-569. The following are a few recent papers which are directly related to FeFe hydrogenases and thus are worth citing:

- a) Pelmeshnikov et al, J. Am. Chem. Soc., 2017, 139 (46), pp16894–16902
- b) Cramer et al, <https://doi.org/10.1002/anie.201805144>
- c) Pelmeshnikov et al, <https://doi.org/10.1002/anie.201804601>

Answer:

The study (Pelmeshnikov et al, J. Am. Chem. Soc., 2017, 139 (46), pp 16894–16902) has already been cited by us but we included the study by Cramer and coauthors (doi.org/10.1002/anie.201805144) as suggested by reviewer 4.

REVIEWERS' COMMENTS:

Answer:

We thank Reviewers 1-4 for their encouragement, insightful contributions and suggestions and are happy about their final recommendation.

Reviewer #1 (Remarks to the Author):

This is an excellent contribution. It appears that all concerns of 4 reviewers were seriously and conscientiously addressed. I see no reason to hold up this publication. It is recommended with enthusiasm.

Reviewer #2 (Remarks to the Author):

The manuscript has been revised in a very satisfactory way and many additional information are now enriching further the nice and accurate work proposed. It is to appreciate the fact that the authors, upon the suggestions given, have increased the level of detail to the benefit of the whole scientific community. To further go in the same direction I would also recommend citing a few missing references to complete the recognition of (and give to readers the direct link to) other works done on proton transfer mutants (Cornish AJ, et al. Biochemistry. 2016 Jun 7;55(22):3165-73., Morra S et al., Biochim Biophys Acta. 2016 Jan;1857(1):98-106., Ratzloff MW et al., J Am Chem Soc. 2018 Jun 20;140(24):7623-7628.)

Answer:

We added these literature sources at appropriate positions either in the main text or in the Supplementary Discussion part.

Reviewer #3 (Remarks to the Author):

I thank the authors for their clear responses to my questions and the resultant modifications to the manuscript. I am happy to recommend publication in its current form

Reviewer #4 (Remarks to the Author):

The revisions are fine with me. I understand some of the suggested context might be too large to be include in the main text but the one included in the main text in this version does make the manuscript stronger. The reconstruction of Fig. 3 makes the work more impressive. I propose that the manuscript be accepted.

Answer:

We thank reviewer 4 for her/his understanding concerning this aspect.